# Learning Reward and Policy Jointly from Demonstration and Preference Improves Alignment

**Chenliang Li**[1], **Siliang Zeng**[2], **Zeyi Liao**[3], **Jiaxiang Li**[2], **Dongyeop Kang**[2],
**Alfredo Garcia**[1], **Mingyi Hong**[2]

[1]Texas A&M University, College Station, [2]University of Minnesota, Twin Cities,
[3]The Ohio State University, Columbus

## Abstract

Aligning to human preferences and/or intentions is an important requirement for contemporary foundation models. To ensure alignment, popular approaches such as reinforcement learning with human feedback (RLHF) break down the task into three stages: (i) a model is computed with supervised fine-tuning (SFT) based upon large demonstrations data, (ii) a reward model (RM) is estimated based upon human feedback data, and (iii) reinforcement learning (RL) is used to further refine the SFT model by optimizing the estimated reward model. Demonstrations and human feedback data reflect human user preferences in different ways. As a result, the reward model estimate obtained from *only* human feedback data is likely not as accurate as a reward model estimate obtained from *both* demonstration and human feedback data. A policy model that optimizes the reward model estimate obtained from *both* demonstration and human feedback data will likely exhibit better alignment performance. We introduce a tractable algorithm for finding the reward and policy models and provide a finite-time performance guarantee. Additionally, we demonstrate the efficiency of the proposed solution with extensive experiments including alignment problems in LLMs and robotic control problems in MuJoCo. We observe that the proposed solutions outperform the existing alignment algorithm by large margins, especially when the amounts of demonstration and preference data are unbalanced.

## 1 Introduction

As ChatGPT has taken the world by storm, it is clear that AI systems will soon become ubiquitous in our lives. For instance, Large Language Models (LLMs) have been used to solve hard problems including video gaming (Berner et al., 2019; Mnih et al., 2015), autonomous control (Bellemare et al., 2020), and robotic manipulation (Kalashnikov et al., 2018; Kober & Peters, 2008). In this context, the notion of *alignment* plays an increasingly important role in the design and training of AI systems. Loosely speaking, alignment refers to the performance guarantee that the AI system will generate outcomes that are intended or preferred by the human user without undesirable side effects or behaviors such as deception (Park et al., 2023) or manipulation (Perez et al., 2022). As human user intentions or preferences may vary under specific contexts, it is critical that the AI system adapts to evolving user preferences and/or intentions (Leike et al., 2018).

The alignment problem is a learning problem with (at least) three types of input data: the demonstration data (consists of prompts and human-generated continuations), the preference data (consists of prompts and pairs of human-ranked responses), as well as prompts without any responses. Moreover, the process of aligning an LLM model is typically undertaken in successive stages. For example, the well-known RLHF approach adopted by Ouyang et al. (2022) starts with a supervised fine-tuning model (SFT) followed by reward model (RM) estimation based upon human-labeled preference data. The process closes with a final alignment stage in which reinforcement learning (RL) is used to optimize the estimated reward model. Similar strategies have been used in other related works such as Rafailov et al. (2023); Li et al. (2023); Zhu et al. (2023); Liu et al. (2023). The

approach to alignment based on successive stages may facilitate computation, but it is at the expense of inefficient exploitation of data. To illustrate, consider the three-stage RLHF approach proposed in Ouyang et al. (2022), in the extreme case where the amount of high-quality preference data is quite limited, the reward model trained cannot adequately reflect the preferences of the human, which may lead to unsatisfactory performance in the RL stage. Further, the reward model estimate obtained from *only* the preference data fails to exploit the information about human users' preferences that are implicit in demonstration data. It is therefore reasonable to expect that a policy model that is fine-tuned with the reward model estimate obtained from *both* demonstration and human feedback may exhibit better alignment performance.

An alternative to the successive approach to alignment consists of *jointly* training the reward and policy models by leveraging demonstration and preference data. In contrast to the successive approach adopted in most of the current alignment approaches, the joint approach to reward and policy learning makes use of all available data, hence mitigating the risk of optimizing an inaccurate reward model. However, a joint approach to learning reward and policy models may improve alignment at the expense of potentially significant additional computational effort.

**Contribution.** We introduce an algorithm jointly learning reward and policy models named Alignment with Integrated Human Feedback (AIHF) with a finite-time performance guarantee. This approach leverages recent advances in Inverse Reinforcement Learning (IRL) (Arora & Doshi, 2021; Zeng et al., 2022b), stochastic choice theory (Blavatskyy & Pogrebna, 2010) and bi-level optimization (Hong et al., 2020; Ji et al., 2021; Khanduri et al., 2021). The proposed formulation integrates SFT, RM, and RL into a single stage, so that reward modeling and policy optimization can *fully* leverage all the available human feedback data. More specifically, in the proposed algorithm, the policy is updated to improve alignment with the current reward model estimate and the reward model is updated to improve the fit to demonstration and human feedback data. As a result, upon convergence, the resulting reward and policy models are *consistent* in the sense that (i) the policy model is optimal with respect to the reward model and (ii) the reward model maximizes the fit to both demonstration and human feedback data. Several existing alignment schemes, such as RLHF (Ouyang et al., 2022) and DPO (Rafailov et al., 2023) and some of their extensions can be seen as particular instances of the proposed formulation. We provide ample empirical evidence that the proposed AIHF solution outperforms the existing alignment algorithms by large margins, especially when the data is *unbalanced*, where the quality and/or quantity of one data category is worse/smaller than that of the other.

## 2 PRELIMINARIES AND RELATED WORK

### 2.1 NOTATION

**The Finite-Horizon MDP Model.** A Markov decision process (MDP) is the tuple $(\mathcal{S}, \mathcal{A}, P, \rho, r, \gamma)$, wherein $\mathcal{S}$ denotes the state space, $\mathcal{A}$ denotes the action space, $P : \mathcal{S} \times \mathcal{A} \times \mathcal{S} \to [0, 1]$ denotes the transition probabilities, $\rho(\cdot)$ is the initial state distribution, $r : \mathcal{S} \times \mathcal{A} \to \mathbb{R}$ denotes the reward function and $\gamma \in (0, 1)$ denotes the discount factor. For every $s_t \in \mathcal{S}$, a randomized policy $\pi(\cdot|s_t)$ is a probability distribution in $\Delta_{|\mathcal{A}|}$, the unit simplex in $\mathbb{R}^{|A|}$. Define $\tau := \{(s_t, a_t)\}_{t=1}^T$ as a (finite horizon $T$) trajectory of state and action pairs. Let $\mathcal{H}_T \subset \prod_{t=1}^T (\mathcal{S} \times \mathcal{A})$ denote all feasible state/action sequence of length $T$.

**MDP Model of LLM.** The generation of text by a language model can be seen as sampling from policies in an MDP model. Specifically, each state $s_t = (x, y_{1:t-1})$ includes the prompt $x$ and all response tokens produced up to that point $y_{1:t-1}$. Each action $a_t = y_t$ represents a token from the vocabulary. The transition kernel $P$ is deterministic, i.e. given tokens $s_t = (x, y_{1:t-1})$ and $a_t = y_t$, the environment will transition to $s_{t+1} = (x, y_{1:t})$. An LLM can be seen as a policy $\pi(\cdot|s_t)$ so that a response of length $T > 0$ to prompt $x$ is obtained with probability: $\pi(y_{1:T}|x) := \prod_{i=1}^T \pi(y_i|x, y_{1:i-1})$

**Human Feedback Data.** Let $\tau := (y_{1:T}, x)$ denote a finite text produced in response to prompt $x$. For a pair of sequences $(\tau_l, \tau_w)$ (which we assume of the same length $T$ for ease of exposition) we write $\tau_l \prec \tau_w$ to indicate the sequence $\tau_w$ is preferred over the sequence $\tau_l$. Following the Bradley-Terry-Luce (BTL) model (Bradley & Terry, 1952), the distribution of preferences over pairs $(\tau_l, \tau_w)$ can be modeled as follows:

$$P(\tau_w \succ \tau_l) = \frac{\exp R(\tau_w; \theta)}{\exp R(\tau_w; \theta) + \exp R(\tau_l; \theta)} = \sigma(R(\tau_w; \theta) - R(\tau_l; \theta)) \quad (1)$$

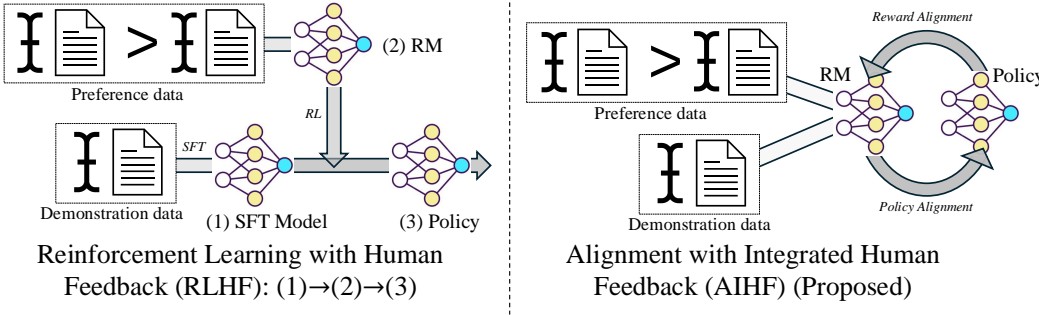

Figure 1: Comparison of the RLHF (left) with the proposed AIHF (right).

where $\sigma$ is the sigmoid function and $R(\tau;\theta) := \sum_{t\geq 1}^{T} \gamma^t r(s_t, a_t; \theta)$ and $r(s_t, a_t; \theta)$ is a reward model parametrized by $\theta \in \mathbb{R}^d$.

## 2.2 THE RLHF PIPELINE

RLHF is a popular technique for fine-tuning AI systems to align with human preferences and values. The RLHF approach proposed in Stiennon et al. (2020); Ouyang et al. (2022) consists of the following three-stage: 1) the **supervised fine-tuning (SFT)** stage, where the demonstration data is used to fine-tune the model in a supervised manner; 2) the **reward modeling (RM)** stage, where the preference data is used to train a reward model; 3) **the reinforcement learning (RL)** stage, where the SFT model is further fine-tuned by running RL using the trained reward model. Specifically, the RLHF pipeline can be formally described as follows:

**Supervised Fine-Tuning (SFT)**: Given a demonstration dataset $\mathcal{D}$ consisting of sequences of the form $\tau = \{(s_t, a_t)\}_{t\geq 0}$ the goal is the find the policy $\pi_{\mathrm{SFT}}(\cdot|s_t)$ that maximizes likelihood, i.e.:

$$\pi_{\mathrm{SFT}} = \arg\max_{\pi} \mathbb{E}_{\tau\sim\mathcal{D}} \Big[ \log \prod_{t\geq 0} \Big( \pi(a_t|s_t) \Big)^{\gamma^t} \Big] \tag{2}$$

**Reward Modeling (RM)**: Based upon a dataset $\mathcal{P}$ of preferences over pairs $(\tau_l, \tau_w)$ the estimation of a reward learning problem can be formulated as the following Bradley-Terry-Luce (BTL) model (Bradley & Terry, 1952) (with $\beta > 0$ a hyper-parameter):

$$\max_{\theta\in\mathbb{R}^d} \ell_{\mathrm{RM}}(\theta) := \mathbb{E}_{(\tau_w\succ\tau_l)\in\mathcal{P}} \Big[ \log \Big( \sigma\big( \frac{1}{\beta}\big( R(\tau_w;\theta) - R(\tau_l;\theta) \big) \big) \Big) \Big]. \tag{3}$$

**Reinforcement Learning (RL)**: Let $\hat{\theta}_{\mathcal{P}}$ denote the solution to problem (3). The last stage in the RLHF development pipeline consists of solving the problem:

$$\pi_{\mathrm{RLHF}} = \arg\max_{\pi} \mathbb{E}_{\tau\sim\pi} \Big[ \sum_{t\geq 0} \gamma^t \big[ r(s_t, a_t; \hat{\theta}_{\mathcal{P}}) - \beta D_{\mathrm{KL}}\big( \pi(\cdot|s_t)\|\pi_{\mathrm{SFT}}(\cdot|s_t) \big) \big] \tag{4}$$

where $D_{\mathrm{KL}}\big( \pi(\cdot|s_t)\|\pi_{\mathrm{SFT}}(\cdot|s_t) \big) := \sum_{a\in\mathcal{A}} \pi(a|s_t) \log \frac{\pi(a|s_t)}{\pi_{\mathrm{SFT}}(a|s_t)}$ is the Kullback-Leibler (KL) divergence, $\pi_{\mathrm{SFT}}$ is the supervised fine-tuning model. Due to the space limit, we put the rest of the literature review in the Appendix A.1.

## 3 ALIGNMENT WITH INTEGRATED HUMAN FEEDBACK (AIHF)

As mentioned before, the reward model obtained in (3) fails to exploit the information about human users' preferences that are implicit in demonstration data. As a result, the fine-tuned model obtained with RLHF may exhibit unsatisfactory alignment performance (this phenomenon will be discussed more concretely in Sec. 3.4). Below we introduce a new approach to jointly train reward and policy models by simultaneously leveraging demonstration and human feedback data.

### 3.1 A META-FORMULATION

Towards developing an approach that can model the *entire* alignment process with a common parametrization for both policy and reward models, consider the following *meta*-formulation, termed

Alignment with Integrated Human Feedback (AIHF):

$$(\textbf{AIHF}) \quad \max_{\theta} \quad L(\theta) := w_1 L_1(\pi_\theta) + L_2(R(\cdot; \theta)) \tag{5a}$$

$$\text{s.t.} \quad \pi_\theta := \arg\max_{\pi} L_3(\pi; R(\cdot; \theta)) \tag{5b}$$

where $\theta \in \mathbb{R}^d$ is a parameter; $L_1(\pi_\theta)$ is a measure of fit of the parameterized policy $\pi_\theta$ to demonstration data and $L_2(R(\cdot; \theta))$ is a measure of fit of the parameterized reward model $R(\cdot; \theta)$ to the preference data and $L_3(\pi, R(\cdot; \theta))$ is a measure of performance of policy $\pi$ with respect to reward model $R(\cdot; \theta)$. $w_1 \geq 0$ is one balancing coefficient reflecting the relative size of demonstration versus preference data. Note that in the lower level policy optimization, the optimal policy corresponding to the the reward model $R(\cdot; \theta)$ is actually determined by the reward parameter $\theta$, where we can denote the optimal policy under the the reward model $R(\cdot; \theta)$ as $\pi_{R_\theta}^* := \pi_\theta$. Therefore we simply put $\pi_\theta$ since the optimal policy under a certain reward model $R(\cdot; \theta)$ is actually determined by the reward parameters $\theta$.

See Fig. 1 for an illustration of AIHF. The AIHF (5) is a *meta*-problem that models the alignment problem. It has two levels: an upper-level problem in which the goal is to find policy and reward models that jointly maximize a measure of fit to demonstrations and preference datasets; and a lower-level problem which ensures that the policy model optimizes performance with respect to the reward model. Its components can be customized to yield specific alignment formulations and algorithms. Before diving into various customizations, let us discuss the advantages of this formulation.

**Generality.** One can specialize the loss functions and problem parameters to yield a number of existing alignment formulations. Such generality implies that algorithms developed for (5) are easily applicable to different special formulations it covers. For more details see Sec. 3.3.

**Joint optimization.** The formulation jointly optimizes the reward and the policy. One benefit here is that it can strengthen the reward model through integrating both demonstrations and pairwise comparisons. Compared with the standard RLHF pipeline, through integrating additional data source such as demonstrations to train the reward model, it can further boost the policy optimization subroutine to achieve better alignment performance. See Sec. 4 for a detailed discussion on how the reward parameter $\theta$ is updated by leveraging such demonstration, and see Tab. 4 for the experimental comparison between the reward model learned by RLHF and by our AIHF (5).

**Dataset Integration.** Clearly, the reward learning process leverages all the available data, therefore, we can expect that a high-quality reward model and its induced optimal policy can still be obtained even under unfavorable situations where the preference data is not sufficient. See Sec. 5 for experimental evidences.

### 3.2 SPECIFICATION OF AIHF

In this section, we specify the formulation (5). Let us begin with the choice of $L_1$. It can be directly instantiated by using one objective similar to (2), which is the likelihood function over the collected expert demonstrations. Note that we aim to optimize the reward parameter $\theta$ to align with human feedback in (5a), thus the objective of $L_1$ can be specialized as a maximum likelihood function over expert demonstrations as below:

$$L_1(\pi_\theta) := \mathbb{E}_{\tau \sim \mathcal{D}} \left[ \log \prod_{t \geq 0} \left( \pi_\theta(a_t | s_t) \right)^{\gamma^t} \right] = \mathbb{E}_{\tau \sim \mathcal{D}} \left[ \sum_{t \geq 0} \gamma^t \log \pi_\theta(a_t | s_t) \right]. \tag{6}$$

Here $\pi_\theta$ optimizes the measure of performance $L_3(\pi; R(\cdot; \theta))$ for a reward model $R(\cdot; \theta)$ as:

$$L_3(\pi; R(\cdot; \theta)) := \mathbb{E}_{s_0 \sim \rho, \tau \sim \pi} \left[ R(\tau; \theta) - \beta \sum_{t \geq 0} \gamma^t D_{\text{KL}} \left( \pi(\cdot | s_t) \| \pi^0(\cdot | s_t) \right) \right] \tag{7}$$

where $\pi^0$ is some initial policy and $\beta > 0$ is temperature parameter.

Next, we specify $L_2$. To ensure internal model consistency, we identify the likelihood function for preference data so it is in accordance with the preferences implied by the reward model $R(\cdot; \theta)$ used in the definitions of $L_1$ and $L_3$. Thus, the optimal distribution $\mu_\theta$ over the set of $T$-long sequence of state-action pairs is defined as follows:

$$\mu_\theta := \arg\max_{\mu \in \Delta^T} \mathbb{E}_{\tau \sim \mu} \left[ R(\tau; \theta) - \beta \mathcal{D}_{\text{KL}}(\mu \| \mu^0) \right]$$

where $\Delta_T$ denotes the simplex on $\mathcal{H}_T$ and $\mu^0$ is a prior distribution on the trajectories. It can be shown the solution of the above problem is of the form:

$$\mu_\theta(\tau) = \frac{\mu^0(\tau)\exp\left(R(\tau;\theta)/\beta\right)}{\sum_{\tau'\in\mathcal{H}_T}\mu^0(\tau')\exp\left(R(\tau';\theta)/\beta\right)}.$$

With this result, we can now obtain a model for the likelihood that sequence $\tau_j$ is preferred over $\tau_j$. By the *independence of irrelevant alternatives* property (Fudenberg et al., 2015) of the optimal choice $\mu_\theta$, when the set of feasible choices is reduced from $\mathcal{H}_T$ to just the the two-tuple $\{\tau_l, \tau_w\}$, the likelihood that sequence $\tau_w$ is preferred over $\tau_l$ is given by $\mathbb{P}_\theta(\tau_w \succ \tau_l) := \frac{\mu_\theta(\tau_w)}{\mu_\theta(\tau_l)+\mu_\theta(\tau_w)}$. This motivates the choice of $L_2(\theta)$ as the following *likelihood function*:

$$L_2(R(\cdot;\theta)) = \mathbb{E}_{(\tau_w \succ \tau_l)\in\mathcal{P}}\left[\log\frac{\mu_\theta(\tau_w)}{\mu_\theta(\tau_w)+\mu_\theta(\tau_l)}\right]$$

$$= \mathbb{E}_{(\tau_w \succ \tau_l)\in\mathcal{P}}\left[\log\frac{\mu^0(\tau_w)\exp\left(R(\tau_w;\theta)\right)}{\mu^0(\tau_w)\exp\left(R(\tau_w;\theta)\right)+\mu^0(\tau_l)\exp\left(R(\tau_l;\theta)\right)}\right].$$

With $\mu^0$ equal to the uniform distribution on $\mathcal{H}_T$, this model is equivalent to the BTL model (3):

$$L_2^{\mathrm{BTL}}(\theta) = \ell_{\mathrm{RM}}(\theta) = \mathbb{E}_{(\tau_w \succ \tau_l)\in\mathcal{P}}\left[\log\left(\sigma\left(R(\tau_w;\theta)-R(\tau_l;\theta)\right)\right)\right]. \tag{8}$$

### 3.3 SPECIAL CASES OF AIHF

Next, we discuss how formulation (5) can be specialized to some of the known alignment algorithms.

**Specialization to RLHF-Type Approach.** First, if we set the coefficient $w_1 = 0$ in (5), we obtain:

$$\max_\theta \quad L_2(\theta) \quad \text{s.t.} \quad \pi_\theta := \arg\max_\pi L_3(\pi; R(\cdot;\theta)). \tag{9}$$

Noticed that now the upper- and lower-level problems are completely decomposable, since the upper-level problem solves for the reward parameterization $\theta$, while the lower-level problem solves for the policy (for the given reward), yielding two separate problems, which are exactly the RM and the RL problems in the typical RLHF approach.

**Specialization to DPO-Type Approach.** Consider the relationship between formulation (5) with the DPO-type approaches. Let us set the following objective function $L_1 = \ell_{\mathrm{SFT}}$ and $L_2 = \ell_{\mathrm{RM}}$, and assume that $T = 1$ for the generation process. Relaxing the constraint (5b) which ensures the policy is optimal w.r.t. a certain parameterized model, we can obtain a DPO-type formulation:

$$\max_\pi L(\pi) := w_1 \cdot \mathbb{E}_{\tau^{\mathrm{E}}\sim\pi^{\mathrm{E}}}\left[\log\pi(a^{\mathrm{E}}|s^{\mathrm{E}})\right] + \mathbb{E}_{(\tau_j \succ \tau_i)\sim\pi^P}\left[\log\left(\sigma\left(\beta\log\frac{\pi(a_j|s_j)}{\pi^0(a_i|s_j)}-\beta\log\frac{\pi(a_i|s_i)}{\pi^0(a_j|s_i)}\right)\right)\right]. \tag{10}$$

The above formulation specializes to Liu et al. (2024), which is a slightly generalized version of DPO when *both* demonstration and preference data are used. Setting $w_1 = 0$ reduces to the problem solved by DPO; see Rafailov et al. (2023, Eq. (2)).

**Specialization to Self-Play Approach.** Define $\ell(\cdot)$ as a monotonic and convex loss function, consider setting $L_1 := w_1 \cdot \mathbb{E}_{\tau^E\in\pi^E,\alpha\in\pi(\cdot|s^E)}\ell\left(R(\tau^E;\theta)-R(\tau;\theta)\right)$, and setting $L_2$ and $L_3$ according to (8) and (7), respectively. Note that the choice of $L_1$ means that given demonstration data, we will find a policy which generates trajectories that match the rewards of the demonstration data. Again using DPO type of reformulation, by substituting the reward expression obtained from the optimal policy (42) to $L_1$ and selecting the $\sigma(\cdot)$ as $\ell(\cdot)$, then the AIHF problem in this case becomes:

$$\max_\pi L(\pi) := w_1 \mathbb{E}_{\tau^{\mathrm{E}}\sim\pi^{\mathrm{E}},\tilde{a}\sim\pi(\cdot|s)}\left[\log\left(\sigma\left(\beta\log\frac{\pi(a^{\mathrm{E}}|s^{\mathrm{E}})}{\pi^0(a^{\mathrm{E}}|s^{\mathrm{E}})}-\beta\log\frac{\pi(\tilde{a}|s^{\mathrm{E}})}{\pi^0(\tilde{a}|s^{\mathrm{E}})}\right)\right)\right]$$
$$+ \mathbb{E}_{(\tau_j \succ \tau_i)\sim\pi^P}\left[\log\left(\sigma\left(\beta\log\frac{\pi(a_j|s_j)}{\pi^0(a_j|s_j)}-\beta\log\frac{\pi(a_i|s_i)}{\pi^0(a_i|s_i)}\right)\right)\right]. \tag{11}$$

Note that the first part of the above formulation is similar to what has been proposed in SPIN (Chen et al., 2024), which only utilizes the SFT data.

### 3.4 WHY AIHF CAN OUTPERFORM TWO-STAGE ALIGNMENT APPROACHES?

To understand the difference between the proposed approach and the successive stages approach of the standard alignment pipeline, let us consider the a *static* setting with action set is $A := \{\tau_1, \tau_2, \cdots \tau_N\}$, reward function $R(\cdot) : A \mapsto \mathbb{R}$, and demonstration $\mathcal{D}$ and preference dataset $\mathcal{P}$. In what follows, we compare the optimal solutions for policies obtained by different alignment approaches. Due to space limitation, all derivation in this section is relegated to Appendix A.3.

**Policy with Demonstration Data.** It can be easily shown that when only the demonstration data $\mathcal{D}$ is available, the probability of generating $i$-th data equals to its empirical probability, i.e., $\pi_{\mathrm{SFT}}(\tau_i) = \frac{\#\{\tau_i \text{ in } \mathcal{D}\}}{|\mathcal{D}|}$. Assume that such a policy is parameterized by an *implicit* reward function $R_D$, using the following softmax choice model where $\tau_i \in A$ is selected with probability $\pi_i^*(R) = \frac{\exp(R_i/\beta)}{\sum_{j=1}^N \exp(R_j/\beta)}$ where $R_i := R(\tau_i)$. Assuming a reference value $\widehat{R}_{\mathcal{D}}(\tau_1) = \bar{R}_1$, then the optimal rewards satisfies (See Sec. A.3.1):

$$\frac{\#\{\tau_i \in \mathcal{D}\}}{|\mathcal{D}|} = \pi_i^*(\widehat{R}_{\mathcal{D}}) = \frac{\exp(\widehat{R}_{\mathcal{D}}(\tau_i)/\beta)}{\sum_{j=1}^N \exp(\widehat{R}_{\mathcal{D}}(\tau_j)/\beta)} \quad i \in \{2, \ldots, N\}. \tag{12}$$

This implicit reward will be used shortly to characterize the RLHF policy.

**Policy with Preference Only Data.** Next, it can be shown that when only the preference data $\mathcal{P}$ is available, the reward estimation problem is defined as:

$$\widehat{R}_{\mathcal{P}} = \arg\max_R \ell_{RM}(R) := \mathbb{E}_{(\tau_i \succ \tau_j) \sim \mathcal{P}} \left[ \log \frac{\pi_i^*(R)}{\pi_i^*(R) + \pi_j^*(R)} \right]. \tag{13}$$

Again with a fixed reference value $\widehat{R}_{\mathcal{P}}(\tau_1) = \bar{R}_1$, the solution is (see Sec. A.3.1):

$$\pi_i^*(\widehat{R}_{\mathcal{P}}) = \frac{\sum_{j:j \neq i} |\mathcal{P}_{i \succ j}|}{\sum_{j:j \neq i} |\mathcal{P}_{i,j}| \rho_{-(i,j)}(\pi^*(\widehat{R}_{\mathcal{P}}))} \tag{14}$$

where $|\mathcal{P}_{i \succ j}| := \#\{\tau_i \succ \tau_j \text{ in } \mathcal{P}\}$ and $|\mathcal{P}_{i,j}| := |\mathcal{P}_{i \succ j}| + |\mathcal{P}_{j \succ i}|$ and $\rho_{-(i,j)}(\pi) := \left(1 - \sum_{k \in A \setminus \{i,j\}} \pi_k\right)^{-1}$ is the expected number of times an action *other* than $\tau_i$ or $\tau_j$ is selected when sampling actions from $\pi$ infinitely many times.

**RLHF Policy.** Based on the above results, in Sec. A.3.2 we show that the RLHF approach has the optimal policy $\pi^{\mathrm{RLHF}}(\tau_i) = \pi_i^*\left(\widehat{R}_{\mathcal{D}} + \widehat{R}_{\mathcal{P}}\right)$. That is, the RLHF policy can be seen as the softmax policy for the *sum* of reward estimators obtained from demonstrations and preferences separately.

**AIHF Policy.** Finally, we also find that the AIHF policy is of the form:

$$\pi^{\mathrm{AIHF}}(\tau_i) = \frac{\#\{\tau_i \text{ in } \mathcal{D}\} + \sum_{j \neq i} |\mathcal{P}_{i \succ j}|}{|\mathcal{D}| + \sum_{j \neq i} |\mathcal{P}_{i,j}| \rho_{-(i,j)}\left(\pi^*(\widehat{R}^{\mathrm{AIHF}})\right)}. \tag{15}$$

**Discussion.** Let us summarize our findings. First, $\pi^{\mathrm{RLHF}}$ takes the form of softmax of the *sum* of two rewards, one learned from the SFT stage one learned from reward training stage provides some interesting insight to this popular approach. Second, $\pi^{\mathrm{AIHF}}$ is more robust than $\pi^{\mathrm{RLHF}}$. To see this, suppose that $|\mathcal{D}| \gg |\mathcal{P}|$, i.e. there is more demonstration than preference data. In this case, the policy estimator in (15) will be largely defined by the demonstration data (which is reasonable) whereas the RLHF policy can be noisy since it (soft) maximizes the sum of two reward estimators: one that is more accurate (i.e. the one based on demonstrations, $\widehat{R}_{\mathcal{D}}$) and one that is less accurate (i.e. the one based on preferences $\widehat{R}_{\mathcal{P}}$). A similar observation can be made when $|\mathcal{D}| \ll |\mathcal{P}|$; see Sec. A.3.2 for more details. Third, $\pi^{\mathrm{AIHF}}$ can have less variance as compared with $\pi^{\mathrm{RLHF}}$. Indeed, $\pi^{\mathrm{AIHF}}$ takes a form of a weighted average of the policies estimated separately with demonstration and preference data, as by using (12) and (14), we can re-write (15) as:

$$\pi_i^*(\widehat{R}^{\mathrm{AIHF}}) = \frac{|\mathcal{D}|}{|\mathcal{D}| + \sum_{j \neq i} |\mathcal{P}_{i,j}| \rho_{-(i,j)}\left(\pi^*(\widehat{R}^{\mathrm{AIHF}})\right)} \pi_i^*(\widehat{R}_{\mathcal{D}}) + \frac{\sum_{j \neq i} |\mathcal{P}_{i,j}| \rho_{-(i,j)}\left(\pi^*(\widehat{R}_{\mathcal{P}})\right)}{|\mathcal{D}| + \sum_{j \neq i} |\mathcal{P}_{i,j}| \rho_{-(i,j)}\left(\pi^*(\widehat{R}^{\mathrm{AIHF}})\right)} \pi_i^*(\widehat{R}_{\mathcal{P}})$$

Such averaging entails reduced variance. We also include simple numerical examples in the Appendix A.3.3 to further illustrate this point.

## 4 PROPOSED ALGORITHM FOR AIHF TRAINING

We are now ready to design algorithms for the proposed AIHF formulation (5). To begin with, first note that (5) takes a hierarchical form, and it belongs to the class of problem named *bi-level* optimization, first developed in the 70s (Fiacco & McCormick, 1990), and recently found many applications in machine learning (Wang et al., 2021; Liu et al., 2021; 2022). Generically speaking, bi-level problems are not easy to optimize; more specifically, in (5), the upper-level problem (5a) is a function of *both* the lower-level optimal solution $\pi_\theta$ and the true parameter $\theta$. It follows that a (stochastic) first-order algorithm for $L(\theta)$ involves some (potentially non-trivial) implicit gradient computation which often involves computing the Hessian matrix for the lower-level objective function. Fortunately, as we will show shortly, with some special choices of $L_1$, $L_2$, $L_3$, one can design some simple and very efficient algorithms.

Before we go to details, we note that throughout this section, we assume that we are searching for a good policy $\pi_\theta$ *and* a reward estimate $r(\cdot, \cdot; \theta)$ to align with human feedback, where the policy $\pi_\theta$ is an optimal solution w.r.t. the certain reward estimate $r(\cdot, \cdot; \theta)$ according to the policy optimization problem (5b). Due to such optimal policy constraint w.r.t. one explicit reward estimate, we design an algorithm to solve such a single-stage, bi-level problem which is different from DPO (Rafailov et al., 2023) that simply optimizes the fixed loss function (10) directly.

On a high level, the proposed algorithm alternates between a policy alignment step (which updates $\pi$ with a fixed reward $r(\cdot, \cdot; \theta)$), and a reward alignment step (which updates $\theta$ using a stochastic gradient, a function of the demonstration and preference data). Next, we study these steps in detail.

**Policy Alignment Step.** One can adopt the standard approaches such as the well-known proximal policy optimization (PPO) (Schulman et al., 2017) algorithm to obtain an approximate optimal policy which solves (7). It is worth noting that, when considering $T = 1$, our discussion leading to (42) indicates the optimal policy takes a much simpler form. In this case, it is possible to consider a simpler method than running PPO to obtain the optimal policy. One alternative way is to use a baseline estimated reward value to perform variance reduction (Li et al., 2023), thus reducing the computational complexity.

It is important to note that, the point of the above discussion is that these different choices for solving the policy alignment problem can be incorporated into our overall approach.

**Reward Alignment.** In this step, we use a stochastic gradient-type algorithm to optimize $L(\theta)$. Towards this end, first, observe that

$$\nabla L(\theta) = w_1 \nabla L_1(\pi_\theta) + \nabla L_2(\theta). \tag{16}$$

Clearly, regardless of the choice of $L_2$, $\nabla L_2$ is relatively easy to compute because the objective is directly related to $\theta$ since $L_2(\theta)$ can be regarded as one supervised learning loss and do not involve the optimal policy $\pi_\theta$. In particular, we have the following expressions:

$$\nabla L_2^{\text{BTL}}(\theta) = \mathbb{E}_{(\tau_w \succ \tau_l) \sim \pi^P} \Big[ \nabla_\theta \log \Big( \sigma \big( R(\tau_w; \theta) - R(\tau_l; \theta) \big) \Big) \Big]. \tag{17a}$$

On the contrary, the computation of $\nabla L_1(\pi_\theta)$ is more involved, since $L_1$ depends on $\theta$ *implicitly* through the corresponding optimal policy $\pi_\theta$. Fortunately, the following lemma indicates that this gradient has a simple and intuitive form as well, and the proof can be found in Appendix A.4.2.

**Lemma 4.1** *Suppose that $L_1$ takes the form of the objective (6) for reward learning from demonstrations, and suppose that $L_3$ takes the form (7) with $c(\cdot)$ being the KL-divergence w.r.t. some initial policy $\pi^0$. Then we have the following expression:*

$$\nabla_\theta L_1(\pi_\theta) = \mathbb{E}_{\tau \sim \pi^E, \tau' \sim \pi_\theta} [\nabla_\theta \big( R(\tau; \theta) - R(\tau'; \theta) \big)] \tag{18}$$

*where $\pi_\theta$ is the optimal policy given the reward model parameterized by $\theta$, with the expression (40).*

Intuitively, if the current policy $\pi_\theta$ has not matched $\pi^E$ yet, then the reward should be improved by going towards the direction suggested by the expert trajectories, while *going away* from those generated by the current policy. Similar to the BTL model, from the gradient expression (18), it is clear that the optimization is toward the direction of increasing the gap between the reward of the real samples (demonstrations) and the synthetic ones (model generated continuations).

In practice, a few approximations need to be made to obtain a stochastic gradient of $L_1$. First, similarly, as before, the precise expectation cannot be obtained because the ground truth policy

---

**Algorithm 1:** *Alignment with Integrated Human Feedback (AIHF)*

---

**Input:** Initialize reward parameter $\theta^0$ and policy model $\pi^0$, the stepsize of reward update $\eta$. Let $\mathcal{P}, \mathcal{D}^{\mathrm{E}}$ denote the preference and the demonstration data, respectively.

  **for** Iteration $k = 0, 1, \ldots, K - 1$ **do**

    **Policy Alignment:** Optimizing $L_3$ by RL subroutine, e.g. PPO, to obtain one improved policy $\pi^{k+1}$

    **Data Sample I:** Sample an expert trajectory $\tau \sim \mathcal{D}^{\mathrm{E}}$ and agent trajectory from $\tau' \sim \pi^{k+1}$

    **Data Sample II:** Sample preference pair $(\tau_w \succ \tau_l) \sim \mathcal{P}$

    **Estimating Gradient:** Calculate one gradient estimator $g^k := w_1 g_1^k + g_2^k$ of

    $\nabla_\theta L(\theta) = w_1 \nabla_\theta L_1(\theta) + \nabla_\theta L_2(\theta)$

    **Reward Alignment:** $\theta^{k+1} := \theta^k + \eta g^k$

  **end for**

---

$\pi^{\mathrm{E}}$ is unknown. Denote an offline demonstration dataset as $\mathcal{D}^{\mathrm{E}} := \{\tau\}$, then one can replace the expectations $\mathbb{E}_{\tau \sim \pi^E}$ by $\mathbb{E}_{\tau \sim \mathcal{D}^{\mathrm{E}}}$. Second, in the second expectation in (18), the trajectories $\tau'$ are sampled from $\pi_\theta$, the optimal policy for a fixed reward parameterization by $\theta$. This means that the *policy alignment* step has to identify the optimal policy $\pi_\theta$ first, which, due to limitations such as computational constraints, and non-linear parameterization, is generally not possible. Instead, we propose to sample from the *current* policy $\pi^{k+1}$ obtained from the previous policy optimization step, where index $k$ represents the iteration counter. Following the approximation steps mentioned above, we construct a stochastic estimator $g_k$ to approximate the exact gradient $\nabla L(\theta_k)$ in (16) as follows:

$$
\begin{aligned}
g_k := w_1 g_1^k + g_2^k := & w_1 \big( \nabla_\theta R(\tau_k^E, \theta_k) - \nabla_\theta R(\tau_k^A, \theta_k) \big) + \big( 1 - \sigma(R(\tau_k^W, \theta_k) - R(\tau_k^L, \theta_k)) \big) \\
& \times \big( \nabla_\theta R(\tau_k^W, \theta_k) - \nabla_\theta R(\tau_k^L, \theta_k) \big).
\end{aligned}
\tag{19}
$$

The above two steps are summarized in Algorithm 1. let us remark on the computational complexity of the proposed algorithm. Note that our algorithm is motivated by a class of popular algorithms in bi-level optimization, where the upper-level and lower-level problems are updated alternatingly using stochastic optimization (Hong et al., 2023). We conclude the section by theoretically inspecting the proposed algorithms.

**Theorem 4.1** *Suppose Assumptions 1 - 2 hold. Selecting stepsize $\alpha := \frac{\alpha_0}{K^\sigma}$ for the reward update step (19) where $\alpha_0 > 0$ and $\sigma \in (0, 1)$ are some fixed constants, and $K$ is the total number of iterations to be run by the algorithm. Then the following result holds:*

$$
\frac{1}{K} \sum_{k=0}^{K-1} \mathbb{E} \left[ \big\| \log \pi_{k+1} - \log \pi_{\theta_k} \big\|_\infty \right] = \mathcal{O}(K^{-1}) + \mathcal{O}(K^{-\sigma})
\tag{20a}
$$

$$
\frac{1}{K} \sum_{k=0}^{K-1} \mathbb{E} \left[ \| \nabla L(\theta_k) \|^2 \right] = \mathcal{O}(K^{-\sigma}) + \mathcal{O}(K^{-1+\sigma}) + \mathcal{O}(K^{-1})
\tag{20b}
$$

*where $\big\| \log \pi_{k+1} - \log \pi_{\theta_k} \big\|_\infty := \max_{s \in \mathcal{S}, a \in \mathcal{A}} \big| \log \pi_{k+1}(a|s) - \log \pi_{\theta_k}(a|s) \big|$. In particular, setting $\sigma = 1/2$, then both quantities in (20a) and (20b) converge with the rate $\mathcal{O}(K^{-1/2})$.*

The above theorem shows that Alg. 1 could converge to stationary point if we take a large loop number $K$. Note that details and proofs of the result above are delegated to Appendix A.4.

## 5 EXPERIMENTS

In this section, we provide numerical evaluations of the proposed method (5) (Alg. 1) and its variants (10) and (11), and comparing them with state-of-the-art methods RLHF (Ouyang et al., 2022), DPO (Rafailov et al., 2023), IPO (Calandriello et al., 2024) and SPIN (Chen et al., 2024). Our experiments demonstrate the advantages of the proposed methods in the following aspects: (1) Reward learning from demonstration and preference is the key to improving over standard RLHF. (2) Using demonstration in reward learning could increase model improvement efficiency (w.r.t. the KL divergence violation) (3) AIHF could reduce the effect of distribution mismatch caused by the sequential alignment method, thus break the performance limits of the state-of-the-art methods.

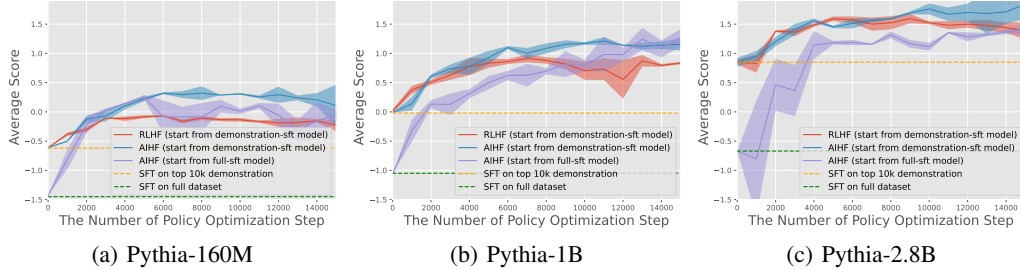

|  |  |  |
|:---:|:---:|:---:|
| (a) Pythia-160M | (b) Pythia-1B | (c) Pythia-2.8B |

Figure 2: Experiment results of Pythia-160M/1B/2.8B policy models, with the reward model trained from Pythia-1.4B. We record the average scores (across three trials) of AIHF and RLHF on the Anthropic-HH test dataset (See Tab. 5 in Appendix for more comparisons with other algorithms)
.

**Models and datasets.** In the first setting, we test Alg. 1 on Anthropic-HH (Bai et al., 2022) dataset[1] with (relatively small) Pythia (Biderman et al., 2023) models[2] as policy models. Anthropic-HH is a preference dataset collected from 52B LLMs that provide two continuations based on helpfulness and harmlessness, and we pick 10k chosen/preferred continuation data to form the demonstration dataset, while others serve as preference dataset and RL prompt dataset. For the HH dataset, We first fine-tune the language models (Pythia-160M/1B/2.8B) through supervised fine-tuning over all chosen responses from the HH dataset for 1 epoch, we call it *full-SFT model* and use it as our base model. Moreover, we also SFT the language model using the selected top 10k chosen responses and name it as *demonstration-SFT model*. For each policy model, we use the exact same model Pythia-1.4B as the reward model.

The other setting we test is on 7B models. We use Ultrafeedback[3] as our preference dataset (61.1k preference data) and Ultrachat200k[4] as the demonstration dataset (208k demonstration data), with mistral-7b-sft-beta [5] (Jiang et al., 2023) as our base model. We use the same mistral-7b-sft-beta model as the initialization of the reward model.

**Evaluation.** For the Anthropic-HH dataset, we present the reward evaluated by the PKU-Alignment/beaver-7b-v3.0-reward model(Ji et al., 2024). In our 7B model experiments, we adopt the widely recognized HuggingFace Open LLM Leaderboard framework (Beeching et al., 2023). This evaluation suite measures LLM performance across six tasks: commonsense reasoning (Arc (Clark et al., 2018), HellaSwag (Zellers et al., 2019), Winogrande (Sakaguchi et al., 2021)), multi-task language understanding (MMLU (Hendrycks et al., 2020)), mimicking human falsehoods (Truth-fulQA (Lin et al., 2021)), and math problem-solving (GSM8K (Cobbe et al., 2021)). Additional implementation details can be found in Appendix A.2.

**Results of small model (1B and 2.8B) experiments.** We observe that the proposed AIHF performs effectively when initiated from both the demonstration-SFT model and the full-SFT model. As shown in Fig. 2, utilizing the same data, AIHF algorithm can eventually outperform RLHF irrespective of the initial model. Furthermore, according to the numerical results as shown in Fig. 3(a), we see that the proposed AIHF algorithm has smaller deviation from the base model compared with the RLHF algorithm. This benefit of the AIHF approach is due to the fact that we incorporate the maximum likelihood IRL objective for both reward learning and policy learning. In this case, both reward model and policy model will be trained to align with the demonstrations, which are also used in the training process of the SFT stage. We also conducted a study on the demonstration/preference data ratio in Fig. 3(b) and 3(c). We observe that AIHF consistently outperforms RLHF across different demonstration/preference data ratio. Furthermore, we also record the performance of the two variants, namely AIHF-DPO (10) and Self-Play-AIHF (11), also RLHF, DPO, IPO and SPIN in Tab. 5. Our proposed AIHF still outperforms all these methods in this experiment setting.

**Results of large model (7B) experiments.** We run AIHF, AIHF-DPO, Self-Play-AIHF along with other methods on the 7B experiment setting. The results are presented in Fig. 4 (the numbers are

---

[1]Dataset available at https://huggingface.co/datasets/Anthropic/hh-rlhf.

[2]Models available at https://huggingface.co/EleutherAI.

[3]Available at https://huggingface.co/datasets/HuggingFaceH4/ultrafeedback_binarized.

[4]Available at https://huggingface.co/datasets/HuggingFaceH4/ultrachat_200k.

[5]Available at https://huggingface.co/HuggingFaceH4/mistral-7b-sft-beta.

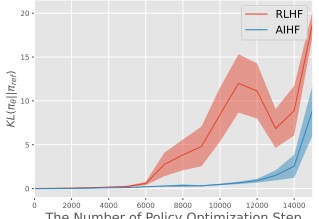 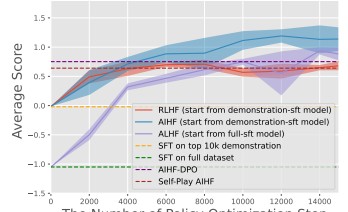 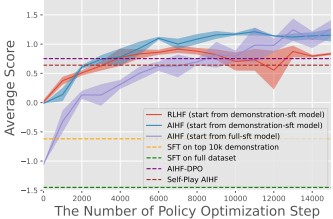

(a) KL divergence to demonstration-SFT policy

(b) AIHF vs RLHF with 10k demonstration, 5k preference

(c) AIHF vs RLHF with 10k demonstration, 10k preference

Figure 3: Experiment results on Pythia-1B policy models, where the reward model is trained from Pythia-1.4B models. We record the average scores of AIHF and RLHF on the Anthropic-HH test dataset, reporting the results across three different trials.

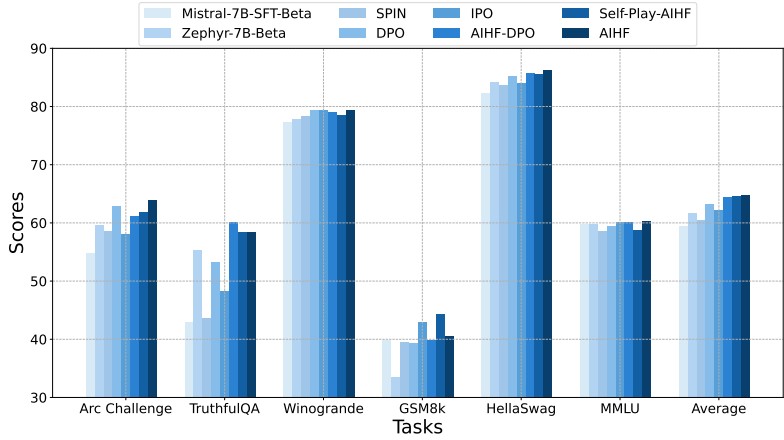

Figure 4: Performance comparison between AIHF-DPO, Self-Play AIHF training across the six benchmark datasets (See also Table 3 in the Appendix).

recorded in Tab. 3), where we can see that similar to the 1B setting, AIHF is consistently outperforming other methods. Additionally, both AIHF-DPO and Self-Play AIHF effectively outperform RLHF model (zephyr-7b-beta). The success of AIHF, as well as Self-Play AIHF and AIHF-DPO further suggests that joint learning from demonstration and preference is indeed beneficial for the alignment. We also conduct ablation study with different choice of $w_1$ in (10), as shown in Tab. 6, the improvement of joint learning methods over baseline is robust. Furthermore,We also evaluate the reward models estimated using different methods (DPO, standard preference learning and AIHF) over the widely used RewardBench (Lambert et al., 2024). The results, illustrated in Tab. 4 in Appendix, show that the reward model trained through the AIHF can achieve significant improvement (especially on reasoning tasks) compared to both standard BTL reward model in (3) and implicit reward model in DPO.

**Other Results.** Due to the page limits, we leave two additional experiments in the appendix: 1) movie review generation with positive sentiment on IMDb dataset (Maas et al., 2011), 2) experiment on Robotics control tasks in MuJoCo (Todorov et al., 2012). For the result of MuJoCo Experiment A.2.1, we observe that even though Behavior Cloning (BC)/SFT could provide a high-performing initialization, RLHF still fails to improve policy quality in the following RL stage. In the contract, AIHF can effectively integrate preferences and demonstrations, leading to a more robust reward function and consequently, a high-quality policy. For the IMDB result (Fig. 6), We show that AIHF is able to alleviate the distribution mismatch between the generated trajectories by the policy, and the data that the learned reward model is able to rank.

## 6 CONCLUSION

In this work, we study the alignment problem when diverse data sources from human feedback are available. Furthermore, we have developed an algorithmic framework that can integrate both expert demonstration and pairwise comparison data from human feedback to learn the reward functions for further guiding policy learning/model fine-tuning in the alignment pipeline. Through extensive evaluations on robotic control tasks and large language model alignment tasks, we demonstrate that our proposed method can outperform existing benchmarks on alignment tasks and is able to recover a better reward model to guide policy learning.

## ACKNOWLEDGMENTS

M. Hong, S. Zeng and J. Li are supported partially by NSF under the grants EPCN-2311007, ECCS-2426064 and CCF-2414372, also by Minnesota Supercomputing Institute. A. Garcia and C. Li are partially supported by W911NF-22-1-0213, ECCS-2240789 and CCF-2414373.

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

## A    APPENDIX

### A.1    LITERATURE REVIEW

#### A.1.1    REWARD LEARNING USING DEMONSTRATION DATA

In the RL literature, a line of work referred to as Inverse Reinforcement Learning (IRL) proposes to *jointly* learn the reward and policy from expert demonstration data. Specifically, the target is to find the parameterized reward function $r(s, a; \theta)$ (resp. an optimal polity $\pi^*(s, a)$) that best explains (resp. mimics) an expert policy $\pi^E$ given the demonstration data $\mathcal{D}$. For example, the well-known maximum entropy IRL (MaxEnt-IRL) framework (Ziebart, 2010; Ziebart et al., 2013; Bloem & Bambos, 2014; Zhou et al., 2017) finds a policy maximizing entropy subject to the expected features that match the empirical averages in the experts observation dataset. However, this approach can only be used to model linear rewards.

Subsequent works such as Levine et al. (2011); Wulfmeier et al. (2015); Zeng et al. (2022b) further improve the MaxEnt-IRL method so that nonlinear reward can be used. For example, Zeng et al. (2022a) proposed a maximum likelihood IRL (ML-IRL) formulation based on the Dynamic Discrete Choice (DDC) model, and a nonlinear reward function is utilized. Additionally, some works Liu & Zhu (2023; 2022) extend IRL to multi-agent settings, broadening its applicability to scenarios involving multiple decision-makers.

It is also shown that when the reward function is linearly parameterized, then the MaxEnt is the Lagrangian dual of the ML-IRL problem (Zeng et al., 2022a). On the other hand, it is worth mentioning that to our best knowledge, almost all IRL-based methods can only utilize demonstration

data, which can be problematic because a large amount of high-quality demonstration data is typically hard to obtain. Further, it is well-known that using demonstration only cannot extract precise human preference, especially in safety-related tasks where the boundaries between permissible and impermissible actions need to be precisely determined; see, e.g., Fischer et al. (2021) which shows that insufficient demonstration dataset could lead to high generalization error.

### A.1.2 JOINT LEARNING FROM DEMONSTRATION AND PREFERENCE

Combining data from demonstrations and human feedback to achieve alignment has also been studied in the robotics literature. In Ibarz et al. (2018), the authors first combine two approaches to learn from human feedback: expert demonstrations and trajectory preferences. The addition of demonstrations to learning from preferences typically results in substantial performance gains compared with using either demonstrations or preferences in isolation. In Palan et al. (2019) and Bıyık et al. (2022), the authors integrate diverse sources of human feedback including demonstrations and pairwise comparisons in a Bayesian approach to learn reward functions that are assumed to be linear in carefully selected features and evaluate their proposed method on robot learning platform. Moreover, their proposed methods need to actively generate preference queries, which are expensive to collect in practical applications. In contrast, the approach proposed in this paper is not Bayesian and does not include the requirement that the reward model is linear in pre-selected features.

### A.1.3 OTHER APPROACHES TO ALIGNMENT

Other approaches to alignment include Direct Preference optimization (DPO) (Rafailov et al., 2023) and Inverse Preference Learning (IPL) (Hejna & Sadigh, 2023) both remove the need for explicit reward modeling, and they directly extract the policy from preferences. This greatly reduced the training complexity, but it has been observed that these algorithms can be unstable in the training process (Azar et al., 2023; Xu et al., 2024). There is also a large number of works that aim to learn reward functions from rating (Daniel et al., 2014) or ranking (Yuan et al., 2023; Myers et al., 2022). Hong et al. (2024) proposed a single-stage supervised learning algorithm ORPO that can perform supervised fine-tuning and preference alignment in one training session without maintaining. However, all of these works highly rely on high-quality human feedback, which is often more difficult and expensive to obtain.

### A.2 EXPERIMENT SETUP AND ADDITIONAL RESULT

### A.2.1 MUJOCO TASKS

In MuJoCo, we consider several robotic control tasks with continuous action space. We evaluate the performance of our proposed algorithm on aligning robot behaviors with provided demonstrations and preference data. After the robot training stage, we leverage the ground-truth reward function from the environment to evaluate the performance.

**Data.** Following the similar data generation pipeline in Brown et al. (2019), we generate the expert demonstrations and preference dataset as follows. We first train an expert agent by leveraging the ground-truth reward function and the popular Soft Actor-Critic (SAC) algorithm Haarnoja et al. (2018), which is developed to solve policy optimization problems with continuous action space. During the training process, we save the policy checkpoints and collect 30k samples from each checkpoint. To achieve precise control of dataset quality, we categorize the data collected into three different classes: low-, medium-, and high-quality datasets according to the performance of the checkpoints. Then we combine the low- and medium-quality data as the preference dataset and use high-quality as demonstration data.

**Results.** We show that AIHF is able to integrate (insufficient amount of) demonstration data and (not-so-high-quality) preference data to generate high-quality policy, and it significantly outperforms the RLHF and IPL. Note that IPL (Hejna & Sadigh, 2024) is actually the DPO-type algorithm applied to multi-horizon MDP, therefore it is an ideal baseline for the MuJoCo setting (since it the underlying problem is a multi-horizon MDP). In Figure 5, we can see that AIHF outperforms both RLHF and IPL algorithms. We also observe that due to the limited number of demonstration data, even BC could provide a high-performing initialization, RLHF still fails to improve policy quality in the following RL stage Ross & Bagnell (2010); Zeng et al. (2022b). Moreover, since the preference data quality is only of low-to-medium quality, the RL step based on the learned reward

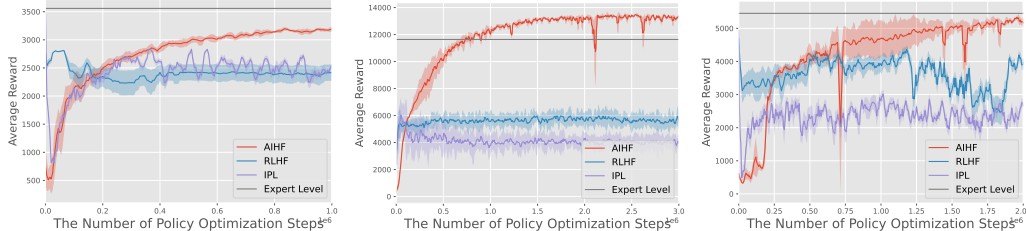

Figure 5: **Top-Left: Hopper Environment; Top-Right: HalfCheetah Environment; Bottom: Walker2d Environment**; AIHF (orange) vs RLHF (blue) vs IPL (purple) (Hejna & Sadigh, 2024); results are averaged over 3 independent runs. We use 10k demonstrations and 20k preferences. The RLHF and IPL curve is initialized from a policy pre-trained by BC; the AIHF from a random policy. The performance is compared against the # of SAC steps performed (for AIHF each policy alignment performs 5k steps of SAC.)

model fails to significantly boost the fine-tuning performance. In contrast, clearly the proposed AIHF can effectively integrate preferences and demonstrations, leading to a more robust reward function and consequently, a high-quality policy.

In SAC, both the policy network and Q network are (64, 64) MLPs with ReLU activation function, and the step size is set to $3 * 10^{-3}$, we parameterize the reward function by a (64, 64) MLPs with ReLU activation function. For the reward network, we use Adam as the optimizer, and the step size is set to be $1 * 10^{-4}$.

The quality of the preference dataset and demonstration dataset are listed in Tab. 1.

| Task \ Dataset | Non-prefer Data | Prefer Data | Demonstration Data |
|---|---|---|---|
| Hopper-v2 | $2345.20 \pm 329.93$ | $3024.63 \pm 40.52$ | $3559.61 \pm 73.12$ |
| HalfCheetah-v2 | $7226.37 \pm 126.88$ | $9434.42 \pm 1315.13$ | $11635.42 \pm 236.51$ |
| Walker2d-v2 | $3952.60 \pm 444.45$ | $5091.71 \pm 291.73$ | $5453.41 \pm 71.07$ |

Table 1: The quality of preference and demonstration.

### A.2.2 SENTIMENT-CONTROLLED GENERATION

**Dataset Generation:** In the IMDb sentiment completion task, we generate the demonstrations and preference datasets using the following procedure. Initially, we train a Language Model by employing the ground-truth reward function DISTILBERT-IMDB and the Proximal Policy Optimization (PPO) algorithm on 30% of the training dataset for IMDb. Throughout the training process, we save the policy checkpoint every 500 PPO steps. Subsequently, we select an additional 40% of the training dataset and generate a response for each prompt for each checkpoint. According to the evaluation score of each generation, we categorize the data collected into different classes: low-, medium-, and high-quality datasets, then we combine low-quality and medium-quality as preference datasets, and use high-quality as demonstration datasets.

**Training:** After acquiring the preference and demonstration datasets, we train the proposed algorithm AIHF and baselines on the remaining 30% of prompts from the training dataset. We evaluate the performance of each algorithm using the test datasets for IMDb and HH, along with their corresponding ground truth reward functions. For the GPU resources, we use $8\times$ A100 40G for all the experiments.

**Results: Policy Quality.** We find that the proposed approach works well when either preference or demonstration data, or both, are limited. From the 7, we see that by using the same amount of data (10k preference, 10k demonstration), AIHF-based algorithms achieve faster convergence than their RLHF and DPO counterparts.

**Results: Distribution Mismatch.** We show that AIHF is able to alleviate the distribution mismatch between the generated trajectories by the policy, and the data that the learned reward model is able to rank. To evaluate the extend of such mismatch, we use the following three steps: (1) use 1k preference, 1k demonstration to train policy and reward model for RLHF and AIHF ; (2) for a given set of prompts from test dataset, use RLHF and AIHF to perform generation; (3) use the trained reward models to rate the generation; (4) compare with the score generated by the ground truth reward LVWERRA/DISTILBERT-IMDB. Fig. 6 illustrates that the reward score distribution produced by AIHF aligns closely with that of the ground truth reward, whereas that generated by RLHF exhibits a poor match. These results show that the reward model learned by AIHF is able to correctly evaluate the generation produced by the final policy.

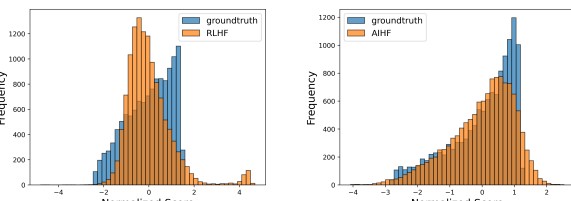

Figure 6: Comparison of the distribution of reward score generated by the trained reward models, and the ground truth reward model. RLHF vs ground truth (left); AIHF vs ground truth (right).

From Fig.7, our proposed algorithm AIHF could obtain higher rewards than baseline methods in the IMDb setting for almost all KL values. Although AIHF might get a low score from the ground truth reward model in the earlier step, AIHF would get a higher reward with more iteration and optimization steps. This indicates that with the mix of demonstration data and preference data, we could prevent the policy from known issues of reward hacking, especially when the policy learned more human-aligned features beyond base models (high KL value). Moreover, AIHF is persistent in the number of preference data, presenting that AIHF could still gain benefit from the limited preference data in more optimization steps as long as the demonstration data is high quality enough.

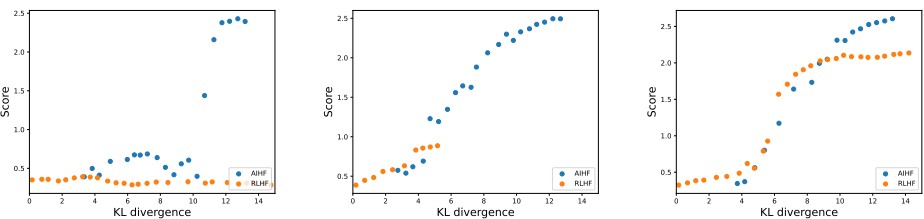

Figure 7: **The frontier of expected reward vs KL to the reference policy in IMDB dataset. fix the demonstration number to 3k** Left: Using 1k preference; Middle: Using 2k preferences; Right: Using 3k preference

### A.2.3 THE RESULT OF 1B AND 2.8B EXPERIMENTS

**Additional Results.** We show here the distribution of the reward values of the continuation generated by the reward models trained by AIHF and RLHF. In Fig. 8, we can observe the distribution of AIHF and RLHF have overlaps in low-quality continuation, however, AIHF can generate more high-quality continuations compared to RLHF, which shows that joint optimization can more effectively align the policy model with the demonstration distribution.

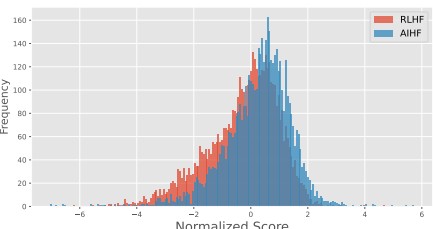 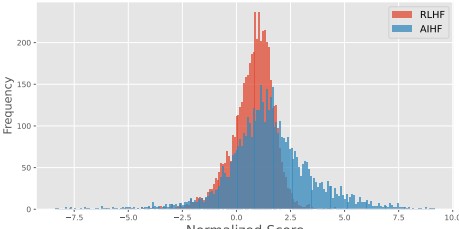

Figure 8: **The Reward Distribution of Helpfulness-controlled Generation. Left: Result on 160m model, Right: Results on 1B model**, This figure reports the reward distribution of generation evaluated by PKU-Alignment/beaver-7b-v3.0-reward for AIHF and RLHF.

We record the performance of the two variants, namely AIHF-DPO (10) and Self-Play-AIHF (11), also RLHF, DPO, IPO and SPIN in Tab. 5. Our proposed AIHF still outperforms all these methods in this experiment setting.

### A.2.4 THE RESULT OF 7B EXPERIMENTS

We utilize DeepSpeed ZeRO-3 (Rajbhandari et al., 2020) to reduce the memory cost, and we use VLLM (Kwon et al., 2023) for accelerate data generation/inference. We use eight NVIDIA A100-40G to do the training with a per-device batch size of 1 for 7B model. We train all models with bfloat16 precision. We set the learning rate to be 5e-7 for the 7B model with the cosine learning rate scheduler, and the max sequence length is set to be 512.

Our device cannot conduct the implementation of the standard AIHF (5) using PPO. Therefore, we employed an online DPO approach (Xiong et al., 2024) for the lower-level policy optimization: 1) at each policy optimization iteration, we first generate multiple continuations from the current policy model over the given prompt dataset, 2) then we utilize our estimated explicit reward models to score each continuation, 3) over each prompt, choose the generated continuation with the highest reward score and the one with lowest reward score as a generated preference pair to run DPO algorithm to fine-tune the policy model. For Self-Play AIHF (11) implementation, we adopt the same strategy as Chen et al. (2024), at each epoch, we generate samples with picked 50k data and generate continuation $\tilde{a} \sim \pi(\cdot|s)$ using the current model $\pi$, then optimize (11) with the sampled $\tilde{a}$.

To evaluate the effectiveness of our approach, we assessed our fine-tuned models on the widely used HuggingFace Open LLM Leaderboard (Beeching et al., 2023) in Table 2. We also list the metric and number of shots used for LLM evaluation on each dataset in Table 3. Moreover, in Tab. 4, we also utilize the Reward-Bench (Lambert et al., 2024) to evaluate the explicit reward models (estimated by standard preference learning in (3) and AIHF) and the implicit reward model (extracted from DPO). The implicit reward is calculated by the policy model similar to what the DPO paper Rafailov et al. (2023) proposed, i.e. $r(a|s) := \log(\frac{\pi(a|s)}{\pi^0(a|s)})$, the first explicit reward is estimated by BTL reward model (3), and second explicit reward is estimated by AIHF. The evaluation results show that AIHF can outperform the benchmark methods in terms of the performance for both policy models and reward models. We also conduct ablation study with different choice of $w_1$ in (10), as shown in Tab. 6, the improvement of joint learning methods over baseline is robust.

| Dataset | Arc Challenge | TruthfulQA MC2 | Winogrande | GSM8K | HellaSwag | MMLU |
|---|---|---|---|---|---|---|
| Metric | acc | acc | acc | strict-match | acc_norm | acc |
| Num. of Shots | 25 | 0 | 5 | 5 | 10 | 5 |

Table 2: A summarization of the benchmarks we use in this work. We list the metric and number of shots used for LLM evaluation on each dataset.

### A.3 WHY AIHF CAN OUTPERFORM TWO-STAGE ALIGNMENT APPROACHES

In this section, we provide detailed analysis for Sec. 3.4.

| Tasks | Arc Challenge | TruthfulQA MC2 | Winogrande | GSM8k | HellaSwag | MMLU | Avg |
|---|---|---|---|---|---|---|---|
| mistral-7b-sft-beta | 54.69% | 42.96% | 77.27% | 39.88% | 82.23% | 59.72% | 59.46% |
| zephyr-7b-beta | 59.64% | 55.18% | 77.82% | 33.51% | 84.19% | 59.76% | 61.68% |
| SPIN | 58.45% | 43.66% | 78.30% | 39.50% | 83.59% | 58.60% | 60.35% |
| DPO | 62.80% | 53.17% | **79.40%** | 39.20% | 85.13% | 59.41% | 63.19% |
| IPO | 58.02% | 48.29% | 79.24% | 42.91% | 83.93% | 60.07% | 62.08% |
| AIHF-DPO | 61.17% | **60.03%** | 79.00% | 39.80% | 85.71% | 60.02% | 64.29% |
| Self-play AIHF | 61.77% | 58.29% | 78.53% | **44.20%** | 85.53% | 58.66% | 64.50% |
| AIHF | **63.90%** | 58.38% | 79.24% | 40.56% | **86.23%** | **60.18%** | **64.75%** |

Table 3: Test performance of AIHF-DPO and Self-Play AIHF based on mistral-7b-sft-beta across HuggingFace Open LLM Leaderboard datasets.

| Reward Model | Chat | Chat Hard | Safety | Reasoning | Average |
|---|---|---|---|---|---|
| DPO Reward Model | 37.43% | 55.92% | 64.14% | 47.33% | 51.21% |
| BTL Reward Model | **95.11%** | **56.58%** | 63.69% | 69.22% | 71.15% |
| AIHF Reward Model | 94.41% | 55.37% | **63.98%** | **76.75%** | **72.63%** |

Table 4: Evaluation of Reward Models in Reward-Bench.

### A.3.1 SFT AND RLHF POLICY

We revisit the RLHF pipeline in the context of a simple softmax choice model where $\tau_i \in A$ is selected with probability

$$\pi_i^*(R) = \frac{\exp(R_i/\beta)}{\sum_{j=1}^N \exp(R_j/\beta)}. \tag{21}$$

For simplicity, we have defined $R_i := R(\tau_i)$. Using this policy model, below let us analyze different policies according to different ways that the rewards are learned.

**Supervised Fine-Tuning (SFT)**: Given a demonstration dataset $\mathcal{D}$ the goal is the find the policy $\pi_{\text{SFT}}$ that maximizes likelihood, i.e.:

$$\pi_{\text{SFT}} := \arg\max_\pi \sum_{\ell=1}^N \mathbb{E}_{\tau_\ell \sim \mathcal{D}} \left[\log \pi_\ell^*(R)\right], \quad \text{s.t.} \sum_{\ell=1}^N \pi_\ell^*(R) = 1. \tag{22}$$

Next let us identify the optimal reward and its corresponding policy in this case. First, let us find the reward function. Write (22) in terms of reward optimization, we have:

$$R^* = \arg\max_R \sum_{\ell=1}^N \mathbb{E}_{\tau_\ell \sim D} \log \frac{\exp(R_\ell/\beta)}{\sum_{j=1}^N \exp(R_j/\beta)} =: L_1(R) \tag{23}$$

The partial derivative of the objective function w.r.t. a component $R_i, i \in [1, \cdots, N]$ is given by:

$$\frac{\partial L_1(R)}{\partial R_i} = \sum_{\ell:\ell \neq i} \mathbb{E}_{\tau_\ell \sim D} \left[ -\frac{1}{\beta} \frac{\exp(R_i/\beta)}{\sum_{j=1}^N \exp(R_j/\beta)} \right] + \mathbb{E}_{\tau_i \sim D} \left[ \frac{1}{\beta} - \frac{1}{\beta} \frac{\exp(R_i/\beta)}{\sum_{j=1}^N \exp(R_j/\beta)} \right]$$

$$= \frac{1}{\beta} \left( \frac{\#\{\tau_i \in \mathcal{D}\}}{|P|} - \sum_\ell \mathbb{E}_{\tau_\ell \in D} \left[ \frac{\exp(R_i/\beta)}{\sum_{j=1}^N \exp(R_j/\beta)} \right] \right), \quad i \in [1:N]. \tag{24}$$

Setting the above partial gradient to zero, we obtain

$$\frac{\#\{\tau_i \in \mathcal{D}\}}{|\mathcal{D}|} = \frac{\exp(\widehat{R}_\mathcal{D}(\tau_i)/\beta)}{\sum_{j=1}^N \exp(\widehat{R}_\mathcal{D}(\tau_j)/\beta)} \quad i \in [1:N], \tag{25}$$

where we used $\widehat{R}_D(\tau_i)$'s to denote the optimal reward estimated from the set of SFT data $\mathcal{D}$. Then according to the definition of the policy given earlier, we obtain

$$\pi_i^*(\widehat{R}_\mathcal{D}) = \frac{\#\{\tau_i \in \mathcal{D}\}}{|\mathcal{D}|}, \ \forall \, i \in [1, \cdots N].$$

| Dataset | DPO | SPIN | IPO | RLHF | AIHF-DPO | Self-Play-AIHF | AIHF |
|---|---|---|---|---|---|---|---|
| 10k Demonstrations, 5k preferences | $0.463 \pm 0.093$ | $0.625 \pm 0.048$ | $0.616 \pm 0.076$ | $0.710 \pm 0.085$ | $0.752 \pm 0.036$ | $0.640 \pm 0.102$ | $1.167 \pm 0.157$ |
| 10k Demonstrations, 10k preferences | $0.474 \pm 0.052$ | $0.625 \pm 0.048$ | $0.650 \pm 0.017$ | $0.896 \pm 0.095$ | $0.798 \pm 0.026$ | $0.674 \pm 0.096$ | $1.190 \pm 0.069$ |

Table 5: Evaluated reward scores of fine-tuned 1B Pythia models in HH dataset, evaluated by PKU-Alignment/beaver-7b-v3.0-reward.

| Tasks | Arc Challenge | TruthfulQA MC2 | Winogrande | GSM8k | HellaSwag | MMLU | Avg |
|---|---|---|---|---|---|---|---|
| `AIHF-DPO`$(w_1 = 0.01)$ | 61.86% | 57.55% | 79.08% | 36.61% | 85.58% | **60.09%** | 63.46% |
| `AIHF-DPO`$(w_1 = 0.001)$ | **63.25%** | 58.73% | **79.16%** | 36.84% | 85.59% | 59.26% | 63.80% |
| `AIHF-DPO`$(w_1 = 0.0001)$ | 61.17% | **60.03%** | 79.00% | **39.80%** | **85.71%** | 60.02% | **64.28%** |

Table 6: Test performance of AIHF-DPO over different choices of the balancing coefficient across HuggingFace Open LLM Leaderboard datasets.

Note that the optimal reward learned from the SFT data $\widehat{R}_D$ should satisfy the system (25). It turns out that there is a unique solution to the following system of equations:

$$\frac{\#\{\tau_i \in \mathcal{D}\}}{|\mathcal{D}|} = \frac{\exp(\widehat{R}_{\mathcal{D}}(\tau_i)/\beta)}{\sum_{j=1}^{N} \exp(\widehat{R}_{\mathcal{D}}(\tau_j)/\beta)} \quad i \in \{2, \ldots, N\} \tag{26}$$

with $\widehat{R}_{\mathcal{D}}(\tau_1) = \bar{R}_1$ a fixed reference value. A simple argument is provided below. The system (26) can be re-written as:

$$\log \frac{\#\{\tau_i \in \mathcal{D}\}}{|\mathcal{D}|} - \log \frac{\#\{\tau_1 \in \mathcal{D}\}}{|\mathcal{D}|} = \frac{1}{\beta}\big(\widehat{R}_{\mathcal{D}}(\tau_i) - \bar{R}_1\big) \quad i \in \{2, \ldots, N\}$$

Hence, rewards are uniquely determined by demonstration data with a fixed reference value $\widehat{R}_{\mathcal{D}}(\tau_1) = \bar{R}_1$.

It is important to note that, despite the fact that in the SFT we may not *directly* learn a reward function, the above analysis says that we are *implicitly* learning a reward $\widehat{R}_D$, assuming that the policy is parameterized as (21). This *implicit* reward will be used later to analyze the RLHF policy. Indeed, if we solve the following SFT *policy optimization* problem directly

$$\max \sum_{\ell} \mathbb{E}_{\tau_\ell \in \mathcal{D}} \log(\pi_\ell), \quad \text{s.t.} \quad \sum_{i \in \mathcal{D}} \pi_i = 1, \tag{27}$$

one can easily obtain that the optimal policy is given by $\pi_i^* = \frac{\#\{\tau_i \in \mathcal{D}\}}{|\mathcal{D}|}, \ \forall \, i \in [1, \cdots N]$.

**Reward Modeling with Preference Data**: Now let us analyze the case where the reward is only learned from a preference dataset. With preference data $\mathcal{P} := \{(\tau_i \succ \tau_j)\}$, the BTL model is:

$$P(\tau_i \succ \tau_j) = \sigma\big(\frac{1}{\beta}\big(R(\tau_i) - R(\tau_j)\big)\big) = \frac{\pi_i^*(R)}{\pi_i^*(R) + \pi_j^*(R)}.$$

The reward estimation problem is defined as:

$$\widehat{R}_{\mathcal{P}} = \arg\max_R L_2(R) := \sum_{\ell, j: \ell \neq j} \mathbb{E}_{(\tau_\ell \succ \tau_j) \sim \mathcal{P}} \left[\log \frac{\pi_\ell^*(R)}{\pi_\ell^*(R) + \pi_j^*(R)}\right] \tag{28}$$

$$= \sum_{\ell, j: \ell \neq j} \mathbb{E}_{(\tau_\ell \succ \tau_j) \sim \mathcal{P}} \left[\log \left(\frac{\exp(R_\ell/\beta)}{\exp(R_\ell/\beta) + \exp(R_j/\beta)}\right)\right] \tag{29}$$

Take the gradient w.r.t. $R_i$, we obtain

$$
\begin{aligned}
\frac{\partial \ell_{RM}(R)}{\partial R_i} &= \sum_{j:j\neq i} \mathbb{E}_{(\tau_i \succ \tau_j)} \left( \frac{1}{\beta} - \frac{1}{\beta}\left( \frac{\exp(R_\ell/\beta)}{\exp(R_\ell/\beta) + \exp(R_j/\beta)} \right) \right) \\
&\quad + \sum_{j:j\neq i} \mathbb{E}_{(\tau_j \succ \tau_i)} \left[ -\frac{1}{\beta}\left( \frac{\exp(R_\ell/\beta)}{\exp(R_\ell/\beta) + \exp(R_j/\beta)} \right) \right] \\
&= \frac{1}{\beta} \sum_{j:j\neq i} \frac{|\mathcal{P}_{i\succ j}|}{|\mathcal{P}|} \left( 1 - \left( \frac{\exp(R_\ell/\beta)}{\exp(R_\ell/\beta) + \exp(R_j/\beta)} \right) \right) \\
&\quad + \frac{1}{\beta} \sum_{j:j\neq i} \frac{|\mathcal{P}_{j\succ i}|}{|\mathcal{P}|} \left( - \left( \frac{\exp(R_\ell/\beta)}{\exp(R_\ell/\beta) + \exp(R_j/\beta)} \right) \right) \\
&= \frac{1}{\beta} \left( \sum_{j:j\neq i} \frac{|\mathcal{P}_{i\succ j}|}{|\mathcal{P}|} - \frac{|\mathcal{P}_{i,j}|}{|\mathcal{P}|} \frac{\exp(R_\ell/\beta)}{\exp(R_\ell/\beta) + \exp(R_j/\beta)} \right) \\
&= \frac{1}{\beta} \sum_{j:j\neq i} \left( \frac{|\mathcal{P}_{i\succ j}|}{|\mathcal{P}_{i,j}|} - \frac{\pi_i^*(R)}{\pi_i^*(R) + \pi_j^*(R)} \right) \frac{|\mathcal{P}_{i,j}|}{|\mathcal{P}|}
\end{aligned}
\tag{30}
$$

where in the above derivation we have defined $|\mathcal{P}_{i\succ j}| := \#\{\tau_i \succ \tau_j \text{ in } \mathcal{P}\}$ and $|\mathcal{P}_{i,j}| := |\mathcal{P}_{i\succ j}| + |\mathcal{P}_{j\succ i}|$. Setting the gradient to zero, we obtain

$$
\sum_{j:j\neq i} \left( \frac{|\mathcal{P}_{i\succ j}|}{|\mathcal{P}_{i,j}|} - \frac{\pi_i^*(\widehat{R}_P)}{\pi_i^*(\widehat{R}_P) + \pi_j^*(\widehat{R}_P)} \right) \frac{|\mathcal{P}_{i,j}|}{|\mathcal{P}|} = 0.
\tag{31}
$$

We shall denote by $\widehat{R}_\mathcal{P}$ as a solution to the above system of equations. The first-order condition (31) can be written in implicit form as:

$$
\pi_i^*(\widehat{R}_P) = \frac{\sum_{j:j\neq i} |\mathcal{P}_{i\succ j}|}{\sum_{j:j\neq i} \frac{|\mathcal{P}_{ij}|}{\pi_i^*(\widehat{R}_P) + \pi_j^*(\widehat{R}_P)}} = \frac{\sum_{j:j\neq i} |\mathcal{P}_{i\succ j}|}{\sum_{j:j\neq i} \frac{|\mathcal{P}_{ij}|}{1 - \sum_{\ell \in A\setminus\{i,j\}} \pi_\ell^*(\widehat{R}_P)}} := \frac{\sum_{j:j\neq i} |\mathcal{P}_{i\succ j}|}{\sum_{j:j\neq i} |\mathcal{P}_{ij}| \rho_{-(i,j)}(\pi^*(\widehat{R}_P))}
\tag{32}
$$

where we have defined $\rho_{-(i,j)}(\pi) := \left( 1 - \sum_{\ell \in A\setminus\{i,j\}} \pi_\ell \right)^{-1}$ as the expected number of times an action *other* than $\tau_i$ or $\tau_j$ is selected when sampling actions from $\pi$ infinitely many times.

**RLHF Policy.** Based on the above two analysis, we are ready to analyze the RLHF policy. The RLHF policy is defined as follows:

$$
\pi^{\text{RLHF}} = \arg\max_{\pi \in \Delta^N} \ \mathbb{E}_{\tau_i \sim \pi} \left[ \widehat{R}_\mathcal{P}(\tau_i) \right] - \beta \text{KL}(\pi || \pi^{\text{SFT}})
$$

where $\widehat{R}_\mathcal{P}$ is the estimator obtained from preference data, $\pi^{\text{SFT}}$ is the SFT model trained with demonstration dataset $\mathcal{D}$, and $\Delta^N$ is the probability simplex.

It can be easily shown that the solution $\pi^{\text{RLHF}}$ is of the form:

$$
\begin{aligned}
\pi^{\text{RLHF}}(\tau_i) &= \frac{\pi^{\text{SFT}}(\tau_i) \exp\left( \frac{1}{\beta} \widehat{R}_\mathcal{P}(\tau_i) \right)}{\sum_{j=1}^N \pi^{\text{SFT}}(\tau_j) \exp\left( \frac{1}{\beta} \widehat{R}_\mathcal{P}(\tau_j) \right)} \\
&= \frac{\exp\left( \frac{1}{\beta} \widehat{R}_\mathcal{D}(\tau_i) \right) \exp\left( \frac{1}{\beta} \widehat{R}_\mathcal{P}(\tau_i) \right)}{\sum_{j=1}^N \exp\left( \frac{1}{\beta} \widehat{R}_\mathcal{D}(\tau_j) \right) \exp\left( \frac{1}{\beta} \widehat{R}_\mathcal{P}(\tau_j) \right)} \quad \text{(using (26))} \\
&= \frac{\exp\left( \frac{1}{\beta} (\widehat{R}_\mathcal{D}(\tau_i) + \widehat{R}_\mathcal{P}(\tau_i)) \right)}{\sum_{j=1}^N \exp\left( \frac{1}{\beta} (\widehat{R}_\mathcal{D}(\tau_j) + \widehat{R}_\mathcal{P}(\tau_j)) \right)} \\
&= \pi_i^*\left( \widehat{R}_\mathcal{D} + \widehat{R}_\mathcal{P} \right).
\end{aligned}
\tag{33}
$$

### A.3.2 THE AIHF POLICY

**The AIHF Policy.** The proposed AIHF estimation problem is

$$\widehat{R}^{\mathrm{AIHF}} = \arg\max_R \ell_{\mathcal{D}+\mathcal{P}}(R) := |\mathcal{D}|L_1(R) + |\mathcal{P}|L_2(R) \tag{34}$$

where $L_1(R) := \mathbb{E}_{\tau_i \sim \mathcal{D}}[\log \pi_i^*(R)]$ and $L_2(R) := \mathbb{E}_{(\tau_j \prec \tau_i) \sim \mathcal{P}}[\log \frac{\pi_i^*(R)}{\pi_i^*(R)+\pi_j^*(R)}]$, and $\pi_i^*(R)$ is given by (21).

Similarly as before, taking gradient w.r.t. $R_i$, and leverage (24) and (30), we obtain

$$\frac{\partial \ell_{\mathcal{D}+\mathcal{P}}(R)}{\partial R_i} = \frac{\#\{\tau_i \text{ in } \mathcal{D}\}}{|\mathcal{D}|}|\mathcal{D}| - \pi_i^*(R)|\mathcal{D}| + \sum_{j:j\neq i}\left(|\mathcal{P}_{i\succ j}| - \sum_{j\neq i}\frac{\pi_i^*(R)}{\pi_i^*(R)+\pi_j^*(R)}|\mathcal{P}_{i,j}|\right) \tag{35}$$

Setting the above condition to zero, we obtain that the AIHF reward should satisfy the following

$$\#\{\tau_i \text{ in } \mathcal{D}\} + \sum_{j:j\neq i}|\mathcal{P}_{i\succ j}| = \pi_i^*(\widehat{R}^{\mathrm{AIHF}})\Big(|\mathcal{D}| + \sum_{j:j\neq i}\frac{|\mathcal{P}_{i,j}|}{\pi_i^*(\widehat{R}^{\mathrm{AIHF}})+\pi_j^*(\widehat{R}^{\mathrm{AIHF}})}\Big)$$

$$= \pi_i^*(\widehat{R}^{\mathrm{AIHF}})\Big(|\mathcal{D}| + \sum_{j:j\neq i}|\mathcal{P}_{i,j}|\rho_{-(i,j)}\big(\pi^*(\widehat{R}^{\mathrm{AIHF}})\big)\Big)$$

Or equivalently,

$$\pi_i^*(\widehat{R}^{\mathrm{AIHF}}) = \frac{\#\{\tau_i \text{ in } \mathcal{D}\} + \sum_{j\neq i}|\mathcal{P}_{i\succ j}|}{|\mathcal{D}| + \sum_{j:j\neq i}|\mathcal{P}_{i,j}|\rho_{-(i,j)}\big(\pi^*(\widehat{R}^{\mathrm{AIHF}})\big)} \tag{36}$$

The system (36) has a unique solution $\widehat{R}^{\mathrm{AIHF}}$ with a fixed reference value $\widehat{R}^{\mathrm{AIHF}}(\tau_1) = \bar{R}_1$.

**Discussion.** Now let us compare the RLHF policy (33) and AIHF policy (36). First, observe that RLHF policy is a function of the *sum* of two reward functions, one learned from the demonstration data and one learned from the preference data, while the AIHF policy takes a more complex form. To have a better understanding about the two policies, let us consider the extreme cases where the SFT and the preference datasets are *unbalanced*, where either $|\mathcal{D}| \gg |\mathcal{P}|$ or $|\mathcal{P}| \gg |\mathcal{D}|$.

- Case $|\mathcal{D}| \gg |\mathcal{P}|$. In this case, $\pi_i^*(\widehat{R}^{\mathrm{AIHF}}) \approx \frac{\#\{\tau_i \text{ in } \mathcal{D}\}}{|\mathcal{D}|} = \pi_i^*(\widehat{R}_{\mathcal{D}})$, so the AIHF policy will ignore the preference dataset, while focusing on the SFT policy; however, for the RLHF policy, one needs to first separately performs the reward estimation, but in this case $\hat{R}_P$ can be very noisy due to lack of data; Then the RLHF policy will also be noisy since it will be influenced by the noisy reward function $\hat{R}_P$.

- Case $|\mathcal{P}| \gg |\mathcal{D}|$. In this case, let's further assume that $\sum_{j:j\neq i}|\mathcal{P}_{i\succ j}| \gg \#\{\tau_i \text{ in } \mathcal{D}\}$, $\forall i$, that is, for each trajectory $\tau_i$, the number of times it appears in the preference data is much larger than its appearance in the demonstration data. In this case, it is easy to see that $\pi_i^*(\widehat{R}^{\mathrm{AIHF}}) \approx \pi_i^*(\widehat{R}_{\mathcal{P}})$, while the RLHF policy will still be noisy as in the first case.

### A.3.3 NUMERICAL EXAMPLES

**Example 1:** With $\beta = 1$ and only two actions $\tau_1$ and $\tau_2$. Since $\rho_{-(1,2)}(\pi) = \rho_{-(2,1)}(\pi) = 1$, it follows from equations (26), (32), (36) that:

$$\pi_1^{\mathrm{AIHF}} := \pi_1^*(\widehat{R}^{\mathrm{AIHF}}) = \frac{\#\{\tau_1 \text{ in } \mathcal{D}\} + \#\{\tau_2 \prec \tau_1 \text{ in } \mathcal{P}\}}{|\mathcal{D}| + |\mathcal{P}|}$$

$$= \frac{|\mathcal{D}|}{|\mathcal{D}| + |\mathcal{P}|}\pi_1^*(\widehat{R}_{\mathcal{D}}) + \frac{|\mathcal{P}|}{|\mathcal{D}| + |\mathcal{P}|}\pi_1^*(\widehat{R}_{\mathcal{P}}).$$

Slightly abusing notations, let $\pi_1^* := \pi_1^*(R^*)$ where $R^*$ is the ground-truth reward. It follows that $\mathrm{Var}(\pi_1^*(\widehat{R}_{\mathcal{D}})) = \frac{\pi_1^*(1-\pi_1^*)}{|\mathcal{D}|}$, $\mathrm{Var}(\pi_1^*(\widehat{R}_{\mathcal{P}})) = \frac{\pi_1^*(1-\pi_1^*)}{|\mathcal{P}|}$ and

$$\mathrm{Var}(\pi_1^{\mathrm{AIHF}}) = \frac{\pi_1^*(1-\pi_1^*)}{|\mathcal{D}| + |\mathcal{P}|} < \min\{\mathrm{Var}(\pi_1^*(\widehat{R}_{\mathcal{D}})), \mathrm{Var}(\pi_1^*(\widehat{R}_{\mathcal{P}}))\}$$

Hence, the AIHF policy estimate has less variance than either the policy estimated obtained from *only* demonstrations or preferences.

To further illustrate, suppose the ground truth is $R^*(\tau_1) = R^*(\tau_2)$ and in the dataset there are more demonstrations than preferences, i.e. $|\mathcal{D}| \gg |\mathcal{P}|$:

$$\#\{\tau_1 \text{ in } \mathcal{D}\} = \#\{\tau_2 \text{ in } \mathcal{D}\} = 50, \quad \#\{\tau_1 \succ \tau_2 \text{ in } \mathcal{P}\} = 6, \quad \#\{\tau_2 \succ \tau_1 \text{ in } \mathcal{P}\} = 4.$$

With the given data, $\pi_1^{\text{SFT}} = \pi_1^*(\widehat{R}_{\mathcal{D}}) = \frac{\#\{\tau_1 \text{ in } \mathcal{D}\}}{|\mathcal{D}|} = \frac{50}{100}$ and the solution to (32) yields

$$\pi_1^*(\widehat{R}_{\mathcal{P}}) = \frac{\exp \widehat{R}_{\mathcal{P}}(\tau_1)}{\exp \widehat{R}_{\mathcal{P}}(\tau_1) + \exp \widehat{R}_{\mathcal{P}}(\tau_2)} = \frac{6}{10}.$$

Hence, $\pi_1^{\text{AIHF}} = \frac{100}{10+100}\pi_1^*(\widehat{R}_{\mathcal{D}}) + \frac{10}{10+100}\pi_1^*(\widehat{R}_{\mathcal{P}}) = \frac{56}{110}$. It follows from (33) that:

$$\pi_1^{\text{RLHF}} = \frac{\pi_1^{\text{SFT}} \exp \widehat{R}_{\mathcal{P}}(\tau_1)}{\pi_1^{\text{SFT}} \exp \widehat{R}_{\mathcal{P}}(\tau_1) + \pi_2^{\text{SFT}} \exp \widehat{r}_{\mathcal{P}}(\tau_2)}$$

$$= \frac{\exp \widehat{R}_{\mathcal{P}}(\tau_1)}{\exp \widehat{R}_{\mathcal{P}}(\tau_1) + \exp \widehat{R}_{\mathcal{P}}(\tau_2)} = \frac{6}{10}.$$

In this example, the RLHF policy estimator is the furthest away from ground-truth, because it does not correctly use the information provided by the demonstration data which in this case happens by chance to be correct $\pi_1^{\text{SFT}} = \pi_1^*(\widehat{R}_{\mathcal{P}}) = \frac{1}{2}$.

As a second example, again suppose the ground truth is $R^*(\tau_1) = R^*(\tau_2)$ and in the dataset there are more preferences than demonstrations, $|\mathcal{P}| \gg |\mathcal{D}|$:

$$\{\#\tau_1 \text{ in } \mathcal{D}\} = 6, \quad \{\#\tau_2 \text{ in } \mathcal{D}\} = 4, \quad \{\#\tau_1 \succ \tau_2 \text{ in } \mathcal{P}\} = \{\#\tau_2 \succ \tau_1 \text{ in } \mathcal{P}\} = 50. \tag{37}$$

In this case, $\pi_1^{\text{SFT}} = \pi_1^*(\widehat{R}_{\mathcal{D}}) = \frac{6}{10}$ and the solution to (32) yields

$$\pi_1^*(\widehat{R}_{\mathcal{P}}) = \frac{\exp \widehat{R}_{\mathcal{P}}(\tau_1)}{\exp \widehat{R}_{\mathcal{P}}(\tau_1) + \exp \widehat{R}_{\mathcal{P}}(\tau_2)} = \frac{50}{100}.$$

It follows from (33) that:

$$\pi_1^{\text{RLHF}} = \frac{\pi_1^{\text{SFT}} \exp \widehat{R}_{\mathcal{P}}(\tau_1)}{\pi_1^{\text{SFT}} \exp \widehat{R}_{\mathcal{P}}(\tau_1) + \pi_2^{\text{SFT}} \exp \widehat{R}_{\mathcal{P}}(\tau_2)}$$

$$= \frac{\pi_1^{SFT}}{\pi_1^{SFT} + \pi_2^{SFT}} = \frac{6}{10}.$$

Hence, $\pi_1^{\text{AIHF}} = \frac{10}{10+100}\pi_1^*(\widehat{R}_{\mathcal{D}}) + \frac{100}{10+100}\pi_1^*(\widehat{R}_{\mathcal{P}}) = \frac{56}{110}$. In this example, the RLHF policy estimator is again farthest from ground-truth, because it does not correctly dismiss the information provided by the demonstration data which is less informative than preference.

**Example 2:** Let us use an illustrative example to show that RLHF method will result in significant data under-utilization when the demonstration coverage is limited. With $\beta = 1$, assume that there are 50 actions, i.e. $A = \{1, 2, \ldots, 50\}$ and each with a ground-truth reward defined by $R^*(\tau_i) = \frac{1}{\sigma\sqrt{2\pi}} e^{-\frac{(\frac{i}{50} - \mu)^2}{2\sigma^2}}$, where $\mu = 0.5$ and $\sigma = 2$. Assume we can sample demonstration and preference from the ground truth reward distribution: demonstrations are sampled from the multinomial distribution, while preferences are sampled from the BTL model.

In an extreme scenario, let demonstrations only include actions 1 through 45, i.e. $\mathcal{D} \cap \{45, 46, \ldots, 50\} = \varnothing$, while preferences have full coverage across all actions. In the subsequent experiment, we initially sample 2000 demonstrations using the multinomial distribution $\pi_i^* = \frac{\exp R_i^*}{\sum_{j=1}^{j=45} \exp R_j^*}$, and obtain 200 preferences for each preference pair with $P(i \succ j) = \frac{\exp R_i^*}{\exp R_j^* + \exp R_i^*}$. We then calculate the RLHF and AIHF policies as in in (33) and (36) to obtain the result depicted in Figure 9:

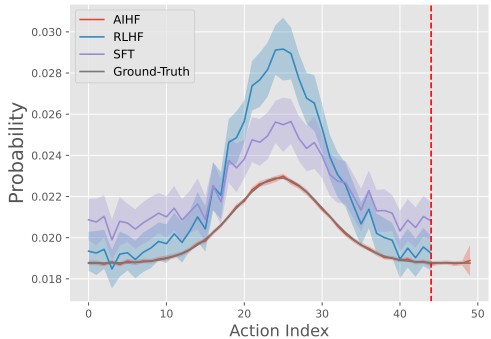

Figure 9: The optimal policy of RLHF, SFT, AIHF, and Ground-truth distribution. The left region of the red dotted line is included in the demonstration, while the right region is uncovered. We report the results with 100 random repeats.

From the result shown in Figure 9, we demonstrate that both SFT and RLHF transfer the weight from uncovered action to covered actions when demonstration coverage is limited, as indicated by $\pi_{SFT}(\tau_i) = 0, \tau_i \in \{45, 46, \ldots, 50\}$. Consequently, the weight of covered actions is significantly higher than the ground truth. However, this issue does not occur when jointly optimizing the demonstration and preference in the AIHF method.

## A.4 PROOFS

### A.4.1 USEFUL LEMMAS

**Assumption 1 (Ergodicity)** *For any policy $\pi$, assume the Markov chain with transition kernel $\mathcal{P}$ is irreducible and aperiodic under policy $\pi$. Then there exist constants $\kappa > 0$ and $\rho \in (0, 1)$ such that*

$$\sup_{s \in \mathcal{S}} \|\mathbb{P}(s_t \in \cdot | s_0 = s, \pi) - \mu_\pi(\cdot)\|_{TV} \leq \kappa \rho^t, \quad \forall\, t \geq 0$$

*where $\| \cdot \|_{TV}$ is the total variation (TV) norm; $\mu_\pi$ is the stationary state distribution under $\pi$.*

Assumption 1 assumes the Markov chain mixes at a geometric rate. It is a common assumption in the literature of RL, which holds for any time-homogeneous Markov chain with finite-state space or any uniformly ergodic Markov chain with general-state space.

**Assumption 2** *For any $s \in \mathcal{S}$, $a \in \mathcal{A}$ and any reward parameter $\theta$, the following holds:*

$$\big\|\nabla_\theta r(s, a; \theta)\big\| \leq L_r, \tag{38a}$$

$$\big\|\nabla_\theta r(s, a; \theta_1) - \nabla_\theta r(s, a; \theta_2)\big\| \leq L_g \|\theta_1 - \theta_2\| \tag{38b}$$

*where $L_r$ and $L_g$ are positive constants.*

2, we next provide the following Lipschitz properties:

**Lemma A.1** *Suppose Assumptions 1 - 2 hold. For any reward parameter $\theta_1$ and $\theta_2$, the following results hold:*

$$|Q^{soft}_{r_{\theta_1}, \pi_{\theta_1}}(s, a) - Q^{soft}_{r_{\theta_2}, \pi_{\theta_2}}(s, a)| \leq L_q \|\theta_1 - \theta_2\|, \quad \forall s \in \mathcal{S}, a \in \mathcal{A} \tag{39a}$$

$$\|\nabla L(\theta_1) - \nabla L(\theta_2)\| \leq L_c \|\theta_1 - \theta_2\| \tag{39b}$$

*where $Q^{soft}_{r_\theta, \pi_\theta}(\cdot, \cdot)$ denotes the soft Q-function under the reward function $r(\cdot, \cdot; \theta)$ and the policy $\pi_\theta$. The positive constants $L_q$ and $L_c$ are defined in Appendix A.4.3.*

### A.4.2 PROOF OF LEMMA 4.1

Before we proceed to the proof of Lemma 4.1, we have the following remark.

**Remark**: The KL-regularized MDP problem described by (5b) and (7) has a closed-form solution:

$$\pi_\theta(a|s) = \frac{\pi^0(a|s)\exp(Q_\theta(s,a)/\beta)}{\sum_{\tilde{a}}\pi^0(\tilde{a}|s)\exp(Q_\theta(s,a)/\beta)}, \tag{40}$$

where the corresponding value function $V_\theta$ and the Q-function $Q_\theta$ are defined as below:

$$V_\theta(s) := \mathbb{E}_{\tau \sim \pi_\theta}\left[R(\tau;\theta) - \beta\sum_{t=0}^{\infty}\gamma^t D_{\mathrm{KL}}\Big(\pi(\cdot|s_t)\|\pi^0(\cdot|s_t)\Big)\Big|s_0 = s\right] \tag{41a}$$

$$Q_\theta(s,a) := r(s,a;\theta) + \gamma\mathbb{E}_{s'\sim P(\cdot|s,a)}\big[V_\theta(s')\big]. \tag{41b}$$

Further, assuming that $T = 1$, i.e., $\tau = (s_0, a_0)$, and considering the LLM alignment problem as a sequence-level training problem (this is a popular simplification in language models, see, e.g., Rafailov et al. (2023)), the closed-form expression of $\pi_\theta$ in (40) can be reduced to:

$$\pi_\theta(a|s) = \frac{\pi^0(a|s)\exp(\frac{1}{\beta}r(s,a;\theta))}{\sum_{a\in A}\left(\pi^0(a|s)\exp(\frac{1}{\beta}r(s,a;\theta))\right)}. \tag{42}$$

Here, under a reward parameter $\theta$ and the corresponding optimal policy $\pi_\theta$ of (40).

Moreover, under a fixed reward parameter $\theta$, we have defined the optimal policy $\pi_\theta$ as below:

$$\pi_\theta := \arg\max_{\pi}\ \mathbb{E}_{\tau^A \sim \pi}\left[\sum_{t=0}^{\infty}\gamma^t\left(r(s_t,a_t;\theta) - \beta\mathcal{D}_{KL}\Big(\pi(\cdot|s_t)\|\pi^0(\cdot|s_t)\Big)\right)\right].$$

According to Uehara et al. (2023), the optimal policy $\pi_\theta$ of (7) has the closed form expression as below:

$$\pi_\theta(a|s) = \frac{\pi^0(a|s)\exp\left(\frac{Q_\theta(s,a;\theta)}{\beta}\right)}{\sum_{\tilde{a}\in\mathcal{A}}\pi^0(\tilde{a}|s)\exp\left(\frac{Q_\theta(s,\tilde{a};\theta)}{\beta}\right)},\quad \forall s\in\mathcal{S}, a\in\mathcal{A}. \tag{43}$$

Based on the closed form of $\pi_\theta$, we can also obtain the closed form of $V_\theta$ as following:

$$V_\theta(s) := \beta\log\left(\sum_{a\in\mathcal{A}}\pi^0(a|s)\exp\left(\frac{Q_\theta(s,a)}{\beta}\right)\right). \tag{44}$$

Then we can re-write the demonstration loss $L_1(\theta)$ as below:

$$L_1(\theta) = \mathbb{E}_{\tau^E \sim \pi^E}\left[\sum_{t=0}^{\infty}\gamma^t\log\pi_\theta(a_t|s_t)\right]$$

$$= \mathbb{E}_{\tau^E \sim \pi^E}\left[\sum_{t=0}^{\infty}\gamma^t\log\left(\frac{\pi^0(a_t|s_t)\exp\left(\frac{Q_\theta(s_t,a_t)}{\beta}\right)}{\sum_{\tilde{a}\in\mathcal{A}}\pi^0(\tilde{a}|s_t)\exp\left(\frac{Q_\theta(s_t,\tilde{a})}{\beta}\right)}\right)\right]$$

$$= \mathbb{E}_{\tau^E \sim \pi^E}\left[\sum_{t=0}^{\infty}\gamma^t\left(\log\left(\pi^0(a_t|s_t)\exp\left(\frac{Q_\theta(s_t,a_t)}{\beta}\right)\right) - \log\left(\sum_{\tilde{a}\in\mathcal{A}}\pi^0(\tilde{a}|s_t)\exp\left(\frac{Q_\theta(s_t,\tilde{a})}{\beta}\right)\right)\right)\right]$$

$$= \mathbb{E}_{\tau^E \sim \pi^E}\left[\sum_{t=0}^{\infty}\gamma^t\left(\log\pi^0(a_t|s_t) + \frac{Q_\theta(s_t,a_t)}{\beta} - \log\left(\sum_{\tilde{a}\in\mathcal{A}}\pi^0(\tilde{a}|s_t)\exp\left(\frac{Q_\theta(s_t,\tilde{a})}{\beta}\right)\right)\right)\right]$$

$$= \frac{1}{\beta}\mathbb{E}_{\tau^E \sim \pi^E}\left[\sum_{t=0}^{\infty}\gamma^t\left(\beta\log\pi^0(a_t|s_t) + Q_\theta(s_t,a_t) - \beta\log\left(\sum_{\tilde{a}\in\mathcal{A}}\pi^0(\tilde{a}|s_t)\exp\left(\frac{Q_\theta(s_t,\tilde{a})}{\beta}\right)\right)\right)\right]$$

$$= \frac{1}{\beta}\mathbb{E}_{\tau^E \sim \pi^E}\left[\sum_{t=0}^{\infty}\gamma^t\left(\beta\log\pi^0(a_t|s_t) + Q_\theta(s_t,a_t) - V_\theta(s_t)\right)\right] \tag{45}$$

Then we can take gradient of $L_1(\theta)$ w.r.t. the reward parameter $\theta$, we have the following expression:

$$
\begin{aligned}
\nabla L_1(\theta) &:= \frac{1}{\beta} \mathbb{E}_{\tau^{\mathrm{E}} \sim \pi^{\mathrm{E}}} \left[ \sum_{t=0}^{\infty} \gamma^t \left( \nabla_\theta \beta \log \pi^0(a_t|s_t) + \nabla_\theta Q_\theta(s_t, a_t) - \nabla_\theta V_\theta(s_t) \right) \right] \\
&= \frac{1}{\beta} \mathbb{E}_{\tau^{\mathrm{E}} \sim \pi^{\mathrm{E}}} \left[ \sum_{t=0}^{\infty} \gamma^t \left( \nabla_\theta Q_\theta(s_t, a_t) - \nabla_\theta V_\theta(s_t) \right) \right] \\
&= \frac{1}{\beta} \mathbb{E}_{\tau^{\mathrm{E}} \sim \pi^{\mathrm{E}}} \left[ \sum_{t=0}^{\infty} \gamma^t \left( \nabla_\theta r(s_t, a_t; \theta) + \gamma \nabla_\theta V_\theta(s_{t+1}) - \nabla_\theta V_\theta(s_t) \right) \right] \\
&= \frac{1}{\beta} \mathbb{E}_{\tau^{\mathrm{E}} \sim \pi^{\mathrm{E}}} \left[ \sum_{t=0}^{\infty} \gamma^t \nabla_\theta r(s_t, a_t; \theta) \right] - \frac{1}{\beta} \mathbb{E}_{s_0 \sim \rho} \left[ \nabla_\theta V_\theta(s_0) \right]
\end{aligned}
\tag{46}
$$

In order to calculate the expression of $\nabla L_1(\theta)$, we further derive the expression of $\nabla_\theta V_\theta(s_0)$:

$$
\begin{aligned}
\nabla_\theta V_\theta(s_0) &= \nabla_\theta \left( \beta \log \left( \sum_{a \in \mathcal{A}} \pi^0(a|s_0) \exp \left( \frac{Q_\theta(s_0, a)}{\beta} \right) \right) \right) \\
&= \beta \sum_{a \in \mathcal{A}} \frac{\pi^0(a|s_0) \exp \left( \frac{Q_\theta(s_0,a)}{\beta} \right)}{\sum_{a \in \mathcal{A}} \pi^0(a|s_0) \exp \left( \frac{Q_\theta(s_0,a)}{\beta} \right)} \frac{\nabla_\theta Q_\theta(s, a)}{\beta} \\
&= \mathbb{E}_{a \sim \pi_\theta(\cdot|s_0)} \left[ \nabla_\theta Q_\theta(s_0, a) \right] \\
&= \mathbb{E}_{a_0 \sim \pi_\theta(\cdot|s_0), s_1 \sim P(\cdot|s_0,a_0)} \left[ \nabla_\theta r(s_0, a_0; \theta) + \gamma \nabla_\theta V_\theta(s_1) \right] \\
&= \mathbb{E}_{\tau^{\mathrm{A}} \sim \pi_\theta} \left[ \sum_{t=0}^{\infty} \gamma^t \nabla_\theta r(s_t, a_t; \theta) \mid s_0 \right]
\end{aligned}
\tag{47}
$$

By plugging (47) into (46), we obtain the following expression:

$$
\nabla L_1(\theta) = \frac{1}{\beta} \mathbb{E}_{\tau^{\mathrm{E}} \sim \pi^{\mathrm{E}}} \left[ \sum_{t=0}^{\infty} \gamma^t \nabla_\theta r(s_t, a_t; \theta) \right] - \frac{1}{\beta} \mathbb{E}_{\tau^{\mathrm{A}} \sim \pi_\theta} \left[ \sum_{t=0}^{\infty} \gamma^t \nabla_\theta r(s_t, a_t; \theta) \right]
\tag{48}
$$

### A.4.3 PROOF OF LEMMA A.1

To prove Lemma A.1, we will prove the equality (39a) and the equality (39b) respectively. The constants $L_q$ and $L_c$ in Lemma A.1 has the expression:

$$
L_q := \frac{L_r}{1 - \gamma}, \quad L_c := \frac{2 L_q L_r C_d \sqrt{|\mathcal{S}| \cdot |\mathcal{A}|}}{1 - \gamma} + \frac{2 L_g}{1 - \gamma}.
$$

### A.4.4 PROOF OF INEQUALITY (39A)

In this subsection, we prove the inequality (39a) in Lemma A.1.

We show that $Q_{r_\theta, \pi_\theta}^{\mathrm{soft}}$ has a bounded gradient with respect to any reward parameter $\theta$, then the inequality (39a) holds due to the mean value theorem. According to the soft Bellman equation, we have shown the explicit expression of $\nabla_\theta Q_{r_\theta, \pi_\theta}^{\mathrm{soft}}(s, a)$ for any $s \in \mathcal{S}$ and $a \in \mathcal{A}$. Using this

expression, we have the following series of relations:

$$
\begin{aligned}
\|\nabla_\theta Q^{\text{soft}}_{r_\theta,\pi_\theta}(s,a)\| &= \left\|\mathbb{E}_{a_0\sim\pi_\theta(\cdot|s_0),s_1\sim P(\cdot|s_0,a_0)}\left[\nabla_\theta r(s_0,a_0;\theta)+\gamma\nabla_\theta V_\theta(s_1)\right]\right\| \\
&\overset{(i)}{=} \left\|\mathbb{E}_{\tau\sim\pi_\theta}\left[\sum_{t\geq 0}\gamma^t\nabla_\theta r(s_t,a_t;\theta)\,\Big|(s_0,a_0)=(s,a)\right]\right\| \\
&\overset{(ii)}{\leq} \mathbb{E}_{\tau\sim\pi_\theta}\left[\sum_{t\geq 0}\gamma^t\left\|\nabla_\theta r(s_t,a_t;\theta)\right\|\,\Big|(s_0,a_0)=(s,a)\right] \\
&\overset{(iii)}{\leq} \mathbb{E}_{\tau\sim\pi_\theta}\left[\sum_{t\geq 0}\gamma^t L_r\,\Big|(s_0,a_0)=(s,a)\right] \\
&= \frac{L_r}{1-\gamma}
\end{aligned}
\tag{49}
$$

where (i) is from the equality (47) in the proof of Lemma A.1, (ii) follows Jensen's inequality and (iii) follows the inequality (38a) in Assumption 2. To complete this proof, we use the mean value theorem to show that

$$
|Q^{\text{soft}}_{r_{\theta_1},\pi_{\theta_1}}(s,a)-Q^{\text{soft}}_{r_{\theta_2},\pi_{\theta_2}}(s,a)| \leq \|\max_\theta \nabla_\theta Q^{\text{soft}}_{r_\theta,\pi_\theta}(s,a)\|\cdot\|\theta_1-\theta_2\| \leq L_q\|\theta_1-\theta_2\|
\tag{50}
$$

where the last inequality follows (49) and we denote $L_q := \frac{L_r}{1-\gamma}$. Therefore, we have proved the Lipschitz continuous inequality in (39a).

### A.4.5 PROOF OF INEQUALITY (39B)

In this section, we prove the inequality (39b) in Lemma A.1.

According to Lemma A.1, the gradient $\nabla L_1(\theta)$ is expressed as:

$$
\nabla L_1(\theta) = \mathbb{E}_{\tau\sim\pi^{\text{E}}}\left[\sum_{t\geq 0}\gamma^t\nabla_\theta r(s_t,a_t;\theta)\right]-\mathbb{E}_{\tau\sim\pi_\theta}\left[\sum_{t\geq 0}\gamma^t\nabla_\theta r(s_t,a_t;\theta)\right].
\tag{51}
$$

Using the above relation, we have

$$
\begin{aligned}
&\|\nabla L_1(\theta_1)-\nabla L_1(\theta_2)\| \\
&\overset{(i)}{=} \left\|\left(\mathbb{E}_{\tau\sim\pi^{\text{E}}}\left[\sum_{t\geq 0}\gamma^t\nabla_\theta r(s_t,a_t;\theta_1)\right]-\mathbb{E}_{\tau\sim\pi_{\theta_1}}\left[\sum_{t\geq 0}\gamma^t\nabla_\theta r(s_t,a_t;\theta_1)\right]\right)-\right.\\
&\qquad\left.\left(\mathbb{E}_{\tau\sim\pi^{\text{E}}}\left[\sum_{t\geq 0}\gamma^t\nabla_\theta r(s_t,a_t;\theta_2)\right]-\mathbb{E}_{\tau\sim\pi_{\theta_2}}\left[\sum_{t\geq 0}\gamma^t\nabla_\theta r(s_t,a_t;\theta_2)\right]\right)\right\| \\
&\leq \underbrace{\left\|\mathbb{E}_{\tau\sim\pi^{\text{E}}}\left[\sum_{t\geq 0}\gamma^t\nabla_\theta r(s_t,a_t;\theta_1)\right]-\mathbb{E}_{\tau\sim\pi^{\text{E}}}\left[\sum_{t\geq 0}\gamma^t\nabla_\theta r(s_t,a_t;\theta_2)\right]\right\|}_{:=\text{term A}}+ \\
&\qquad \underbrace{\left\|\mathbb{E}_{\tau\sim\pi_{\theta_1}}\left[\sum_{t\geq 0}\gamma^t\nabla_\theta r(s_t,a_t;\theta_1)\right]-\mathbb{E}_{\tau\sim\pi_{\theta_2}}\left[\sum_{t\geq 0}\gamma^t\nabla_\theta r(s_t,a_t;\theta_2)\right]\right\|}_{:=\text{term B}}
\end{aligned}
\tag{52}
$$

where (i) follows the exact gradient expression in equation (51). Then we separately analyze term A and term B in (52).

For term A, it follows that

$$\left\| \mathbb{E}_{\tau \sim \pi^{\mathrm{E}}}\left[ \sum_{t \geq 0} \gamma^t \nabla_\theta r(s_t, a_t; \theta_1) \right] - \mathbb{E}_{\tau \sim \pi^{\mathrm{E}}}\left[ \sum_{t \geq 0} \gamma^t \nabla_\theta r(s_t, a_t; \theta_2) \right] \right\|$$

$$\overset{(i)}{\leq} \mathbb{E}_{\tau \sim \pi^{\mathrm{E}}}\left[ \sum_{t \geq 0} \gamma^t \left\| \nabla_\theta r(s_t, a_t; \theta_1) - \nabla_\theta r(s_t, a_t; \theta_2) \right\| \right]$$

$$\overset{(ii)}{\leq} \mathbb{E}_{\tau \sim \pi^{\mathrm{E}}}\left[ \sum_{t \geq 0} \gamma^t L_g \|\theta_1 - \theta_2\| \right]$$

$$= \frac{L_g}{1 - \gamma} \|\theta_1 - \theta_2\| \tag{53}$$

where (i) follows Jensen's inequality and (ii) is from (38b) in Assumption 2.

For the term B, it holds that

$$\left\| \mathbb{E}_{\tau \sim \pi_{\theta_1}}\left[ \sum_{t \geq 0} \gamma^t \nabla_\theta r(s_t, a_t; \theta_1) \right] - \mathbb{E}_{\tau \sim \pi_{\theta_2}}\left[ \sum_{t \geq 0} \gamma^t \nabla_\theta r(s_t, a_t; \theta_2) \right] \right\|$$

$$\overset{(i)}{\leq} \left\| \mathbb{E}_{\tau \sim \pi_{\theta_1}}\left[ \sum_{t \geq 0} \gamma^t \nabla_\theta r(s_t, a_t; \theta_1) \right] - \mathbb{E}_{\tau \sim \pi_{\theta_2}}\left[ \sum_{t \geq 0} \gamma^t \nabla_\theta r(s_t, a_t; \theta_1) \right] \right\|$$

$$+ \left\| \mathbb{E}_{\tau \sim \pi_{\theta_2}}\left[ \sum_{t \geq 0} \gamma^t \nabla_\theta r(s_t, a_t; \theta_1) \right] - \mathbb{E}_{\tau \sim \pi_{\theta_2}}\left[ \sum_{t \geq 0} \gamma^t \nabla_\theta r(s_t, a_t; \theta_2) \right] \right\|$$

$$\overset{(ii)}{\leq} \frac{1}{1 - \gamma} \left\| \mathbb{E}_{(s,a) \sim d(\cdot, \cdot; \pi_{\theta_1})}\left[ \nabla_\theta r(s_t, a_t; \theta_1) \right] - \mathbb{E}_{(s,a) \sim d(\cdot, \cdot; \pi_{\theta_2})}\left[ \nabla_\theta r(s_t, a_t; \theta_1) \right] \right\|$$

$$+ \mathbb{E}_{\tau \sim \pi_{\theta_2}}\left[ \sum_{t \geq 0} \gamma^t \left\| \nabla_\theta r(s_t, a_t; \theta_1) - \nabla_\theta r(s_t, a_t; \theta_2) \right\| \right]$$

$$\overset{(iii)}{\leq} \frac{1}{1 - \gamma} \left\| \sum_{s \in \mathcal{S}, a \in \mathcal{A}} \nabla_\theta r(s_t, a_t; \theta_1) \Big( d(s, a; \pi_{\theta_1}) - d(s, a; \pi_{\theta_2}) \Big) \right\| + \mathbb{E}_{\tau \sim \pi_{\theta_2}}\left[ \sum_{k \geq 0} \gamma^k L_g \|\theta_1 - \theta_2\| \right]$$

$$\overset{(iv)}{\leq} \frac{2 L_r}{1 - \gamma} \|d(\cdot, \cdot; \pi_{\theta_1}) - d(\cdot, \cdot; \pi_{\theta_2})\|_{TV} + \frac{L_g}{1 - \gamma} \|\theta_1 - \theta_2\| \tag{54}$$

where (i) follows the triangle inequality, (ii) is from Jensen's inequality and the definition of the discounted state-action visitation measure $d(s, a; \pi) := (1 - \gamma)\pi(a|s)\sum_{t \geq 0} \gamma^t \mathcal{P}^\pi(s_t = s | s_0 \sim \eta)$; (iii) is from (38b) in Assumption 2;(iv) is from (38a) and the definition of the total variation norm.

Consider the $L_2$ term:

$$L_2(\theta) := \mathbb{E}_{(\tau_i, \tau_w) \sim \pi^P}\left[ \log\left( \sigma\left( R(\tau_w; \theta) - R(\tau_i; \theta) \right) \right) \right]$$

where $\sigma(x)$ is sigmoid function defined by: $\sigma(x) = \frac{1}{1 + e^{-x}}$. We have

$$\nabla_\theta L_2(\theta) = \mathbb{E}_{(\tau_i, \tau_w) \sim \pi^P}\left[ (1 - \sigma(R(\tau_w; \theta) - R(\tau_i; \theta))) \cdot (\nabla_\theta R(\tau_w; \theta) - \nabla_\theta R(\tau_i; \theta)) \right].$$

$$= \mathbb{E}_{(\tau_i, \tau_w) \sim \pi^P}\left[ (\nabla_\theta R(\tau_w; \theta) - \nabla_\theta R(\tau_i; \theta)) - \sigma(R(\tau_w; \theta) - R(\tau_i; \theta))(\nabla_\theta R(\tau_w; \theta) - \nabla_\theta R(\tau_i; \theta)) \right]$$

Using the triangle inequality, we obtain the following equation:

$$\|\nabla L_2(\tau_w, \tau_l; \theta_1) - \nabla L_2(\tau_w, \tau_l; \theta_2)\|$$

$$\leq \underbrace{\left\| \mathbb{E}_{(\tau_i, \tau_w) \sim \pi^P}\left[ (\nabla_\theta R(\tau_w; \theta_1) - \nabla_\theta R(\tau_i; \theta_1)) - (\nabla_\theta R(\tau_w; \theta_2) - \nabla_\theta R(\tau_i; \theta_2)) \right] \right\|}_{:= \text{term A}}$$

$$+ \underbrace{\left\| \mathbb{E}_{(\tau_i, \tau_w) \sim \pi^P}\left[ \sigma(R(\tau_w; \theta_1) - R(\tau_i; \theta_1))(\nabla_\theta R(\tau_w; \theta_1) - \nabla_\theta R(\tau_i; \theta_1)) \right. \right.}$$

$$\left. \left. - \sigma(R(\tau_w; \theta_2) - R(\tau_i; \theta_2))(\nabla_\theta R(\tau_w; \theta_2) - \nabla_\theta R(\tau_i; \theta_2)) \right] \right\|}_{:= \text{term B}} \tag{55}$$

First we bound the term A of (55)

$$\text{term A} = \left\| \left( \left[ \sum_{t \geq 0} \gamma^t \nabla_\theta r(s_t^w, a_t^w; \theta_1) - \gamma^t \nabla_\theta r(s_t^l, a_t^l; \theta_1) \right] \right) - \left( \left[ \sum_{t \geq 0} \gamma^t \nabla_\theta r(s_t^w, a_t^w; \theta_2) - \gamma^t \nabla_\theta r(s_t^l, a_t^l; \theta_2) \right] \right) \right\|$$

$$\leq \left\| \sum_{t \geq 0} \gamma^t \nabla_\theta r(s_t^w, a_t^w; \theta_1) - \gamma^t \nabla_\theta r(s_t^w, a_t^w; \theta_2) \right\| + \left\| \sum_{t \geq 0} \gamma^t \nabla_\theta r(s_t^l, a_t^l; \theta_1) - \gamma^t \nabla_\theta r(s_t^l, a_t^l; \theta_2) \right\|$$

$$\leq \frac{2L_g}{1 - \gamma} \|\theta_1 - \theta_2\| \tag{56}$$

Then we bounded term B of (55):

term B =

$$= \left\| \sigma\left( \left[ \sum_{t \geq 0} \gamma^t r(s_t^w, a_t^w; \theta_1) - \gamma^t r(s_t^l, a_t^l; \theta_1) \right] \right) \left( \left[ \sum_{t \geq 0} \gamma^t \nabla_\theta r(s_t^w, a_t^w; \theta_1) - \gamma^t \nabla_\theta r(s_t^l, a_t^l; \theta_1) \right] \right) \right.$$

$$\left. - \sigma\left( \left[ \sum_{t \geq 0} \gamma^t r(s_t^w, a_t^w; \theta_2) - \gamma^t r(s_t^l, a_t^l; \theta_2) \right] \right) \left( \left[ \sum_{t \geq 0} \gamma^t \nabla_\theta r(s_t^w, a_t^w; \theta_2) - \gamma^t \nabla_\theta r(s_t^l, a_t^l; \theta_2) \right] \right) \right\|$$

$$= \left\| \sigma\left( \sum_{t \geq 0} \gamma^t r(s_t^w, a_t^w; \theta_1) - \gamma^t r(s_t^l, a_t^l; \theta_1) \right) \left( \sum_{t \geq 0} \gamma^t \nabla_\theta r(s_t^w, a_t^w; \theta_1) - \gamma^t \nabla_\theta r(s_t^l, a_t^l; \theta_1) \right) \right.$$

$$- \sigma\left( \sum_{t \geq 0} \gamma^t r(s_t^w, a_t^w; \theta_1) - \gamma^t r(s_t^l, a_t^l; \theta_1) \right) \left( \sum_{t \geq 0} \gamma^t \nabla_\theta r(s_t^w, a_t^w; \theta_2) - \gamma^t \nabla_\theta r(s_t^l, a_t^l; \theta_2) \right)$$

$$+ \sigma\left( \sum_{t \geq 0} \gamma^t r(s_t^w, a_t^w; \theta_1) - \gamma^t r(s_t^l, a_t^l; \theta_1) \right) \left( \sum_{t \geq 0} \gamma^t \nabla_\theta r(s_t^w, a_t^w; \theta_2) - \gamma^t \nabla_\theta r(s_t^l, a_t^l; \theta_2) \right)$$

$$\left. \sigma\left( \sum_{t \geq 0} \gamma^t r(s_t^w, a_t^w; \theta_2) - \gamma^t r(s_t^l, a_t^l; \theta_2) \right) \left( \sum_{t \geq 0} \gamma^t \nabla_\theta r(s_t^w, a_t^w; \theta_2) - \gamma^t \nabla_\theta r(s_t^l, a_t^l; \theta_2) \right) \right\|$$

$$\leq \left\| \sigma\left( \sum_{t \geq 0} \gamma^t r(s_t^w, a_t^w; \theta_1) - \gamma^t r(s_t^l, a_t^l; \theta_1) \right) \left( \sum_{t \geq 0} \gamma^t \nabla_\theta r(s_t^w, a_t^w; \theta_1) - \gamma^t \nabla_\theta r(s_t^l, a_t^l; \theta_1) \right. \right.$$

$$\left. \left. + \sum_{t \geq 0} \gamma^t \nabla_\theta r(s_t^w, a_t^w; \theta_2) - \gamma^t \nabla_\theta r(s_t^l, a_t^l; \theta_2) \right) \right\| + \left\| \left( \sum_{t \geq 0} \gamma^t \nabla_\theta r(s_t^w, a_t^w; \theta_1) - \gamma^t \nabla_\theta r(s_t^l, a_t^l; \theta_1) \right) \right.$$

$$\left. \left[ \sigma\left( \sum_{t \geq 0} \gamma^t \nabla_\theta r(s_t^w, a_t^w; \theta_1) - \gamma^t \nabla_\theta r(s_t^l, a_t^l; \theta_1) \right) - \sigma\left( \sum_{t \geq 0} \gamma^t \nabla_\theta r(s_t^w, a_t^w; \theta_1) - \gamma^t \nabla_\theta r(s_t^l, a_t^l; \theta_1) \right) \right] \right\|$$

$$\leq \frac{2L_g}{1 - \gamma} \|\theta_1 - \theta_2\| + \frac{L_g}{1 - \gamma} \|\theta_1 - \theta_2\| = \frac{3L_g}{1 - \gamma} \|\theta_1 - \theta_2\| \tag{57}$$

Plugging the inequalities (53), (54) to (52), it holds that

$$\|\nabla L(\theta_1) - \nabla L(\theta_2)\|$$

$$\leq \frac{2L_r}{1 - \gamma} \|d(\cdot, \cdot; \pi_{\theta_1}) - d(\cdot, \cdot; \pi_{\theta_2})\|_{TV} + \frac{6L_g}{1 - \gamma} \|\theta_1 - \theta_2\|$$

$$\overset{(i)}{\leq} \frac{2L_r C_d}{1 - \gamma} \|Q_{r_{\theta_1}, \pi_{\theta_1}}^{\text{soft}} - Q_{r_{\theta_2}, \pi_{\theta_2}}^{\text{soft}}\| + \frac{6L_g}{1 - \gamma} \|\theta_1 - \theta_2\|$$

$$\overset{(ii)}{\leq} \frac{2L_r C_d \sqrt{|\mathcal{S}| \cdot |\mathcal{A}|}}{1 - \gamma} \|Q_{r_{\theta_1}, \pi_{\theta_1}}^{\text{soft}} - Q_{r_{\theta_2}, \pi_{\theta_2}}^{\text{soft}}\|_\infty + \frac{6L_g}{1 - \gamma} \|\theta_1 - \theta_2\|$$

$$\overset{(iii)}{\leq} \left( \frac{2L_q L_r C_d \sqrt{|\mathcal{S}| \cdot |\mathcal{A}|}}{1 - \gamma} + \frac{6L_g}{1 - \gamma} \right) \|\theta_1 - \theta_2\|. \tag{58}$$

Define the constant $L_c := \frac{2L_q L_r C_d \sqrt{|\mathcal{S}| \cdot |\mathcal{A}|}}{1 - \gamma} + \frac{5L_g}{1 - \gamma}$, we have the following inequality:

$$\|\nabla L(\theta_1) - \nabla L(\theta_2)\| \leq L_c \|\theta_1 - \theta_2\|.$$

Therefore, we complete the proof of the inequality (39b) in Lemma A.1.

## A.5 Proof of Theorem 4.1

In this section, we prove (20a) and (20b) respectively, to show the convergence of the lower-level problem and the upper-level problem.

### A.5.1 Proof of (20a)

In this proof, we first show the convergence of the lower-level variable $\{\pi_k\}_{k \geq 0}$. Recall that we approximate the optimal policy $\pi_{\theta_k}$ by $\pi_{k+1}$ at each iteration $k$. We first analyze the approximation error between $\pi_{\theta_k}$ and $\pi_{k+1}$ as follows. For any $s \in \mathcal{S}$ and $a \in \mathcal{A}$, we have the following relation:

$$
\begin{aligned}
& \big| \log \big( \pi_{k+1}(a|s) \big) - \log \big( \pi_{\theta_k}(a|s) \big) \big| \\
& \stackrel{(i)}{=} \left| \log \left( \frac{\pi^0(a|s) \exp \big( Q^{\text{soft}}_{r_{\theta_k}, \pi_k}(s,a) \big)}{\sum_{\tilde{a}} \exp \pi^0(\tilde{a}|s) \big( Q^{\text{soft}}_{r_{\theta_k}, \pi_k}(s,\tilde{a}) \big)} \right) - \log \left( \frac{\pi^0(a|s) \exp \big( Q^{\text{soft}}_{r_{\theta_k}, \pi_{\theta_k}}(s,a) \big)}{\sum_{\tilde{a}} \pi^0(\tilde{a}|s) \exp \big( Q^{\text{soft}}_{r_{\theta_k}, \pi_{\theta_k}}(s,\tilde{a}) \big)} \right) \right| \\
& \stackrel{(ii)}{\leq} \big| Q^{\text{soft}}_{r_{\theta_k}, \pi_k}(s,a) - Q^{\text{soft}}_{r_{\theta_k}, \pi_{\theta_k}}(s,a) \big| + \left| \log \left( \sum_{\tilde{a}} \pi^0(\tilde{a}|s) \exp \big( Q^{\text{soft}}_{r_{\theta_k}, \pi_k}(s,\tilde{a}) \big) \right) \right. \\
& \qquad \left. - \log \left( \sum_{\tilde{a}} \pi^0(\tilde{a}|s) \exp \big( Q^{\text{soft}}_{r_{\theta_k}, \pi_{\theta_k}}(s,\tilde{a}) \big) \right) \right|
\end{aligned}
\tag{59}
$$

where (i) follows (40); (ii) is by the triangle inequality. We further analyze the second term in (59).

We first denote the operator $\log(\|w \exp(v)\|_1) := \log(\|\sum_{\tilde{a} \in \mathcal{A}} w \exp(v_{\tilde{a}})\|_1)$, where the vector $w, v \in \mathbb{R}^{|\mathcal{A}|}$ and $v = [v_1, v_2, \cdots, v_{|\mathcal{A}|}], w = [w_1, w_2, \cdots, w_{|\mathcal{A}|}]$. Then for any $v', v'' \in \mathbb{R}^{|\mathcal{A}|}$, we have the following relation:

$$
\begin{aligned}
\big| \log \big( \|w' \exp(v')\|_1 \big) - \log \big( \|w'' \exp(v'')\|_1 \big) \big| & \stackrel{(i)}{=} \big\langle v' - v'', \nabla_v \log \big( \|w \exp(v)\|_1 \big) |_{v=v^c} \big\rangle \\
& \leq \|v' - v''\|_\infty \cdot \|\nabla_v \log \big( \|w \exp(v)\|_1 \big) |_{v=v^c} \|_1 \\
& \stackrel{(ii)}{=} \|v' - v''\|_\infty
\end{aligned}
\tag{60}
$$

where (i) follows the mean value theorem and $v_c$ is a convex combination of $v'$ and $v''$; (ii) follows the following equalities:

$$
[\nabla_v \log \big( \|w \exp(v)\|_1 \big)]_i = \frac{w_i \exp(v_i)}{\sum_{1 \leq a \leq |\mathcal{A}|} w_a \exp(v_a)}, \quad \|\nabla_v \log \big( \|w \exp(v)\|_1 \big)\|_1 = 1, \quad \forall v \in \mathbb{R}^{|\mathcal{A}|}.
$$

Through plugging (60) into (59), it holds that

$$
\begin{aligned}
& \big| \log \big( \pi_{k+1}(a|s) \big) - \log \big( \pi_{\theta_k}(a|s) \big) \big| \\
& \leq \big| Q^{\text{soft}}_{r_{\theta_k}, \pi_k}(s,a) - Q^{\text{soft}}_{r_{\theta_k}, \pi_{\theta_k}}(s,a) \big| + \max_{\tilde{a} \in \mathcal{A}} \big| Q^{\text{soft}}_{r_{\theta_k}, \pi_k}(s,\tilde{a}) - Q^{\text{soft}}_{r_{\theta_k}, \pi_{\theta_k}}(s,\tilde{a}) \big|
\end{aligned}
\tag{61}
$$

Taking the infinity norm over $\mathbb{R}^{|\mathcal{S}| \cdot |\mathcal{A}|}$, the following result holds:

$$
\| \log \pi_{k+1} - \log \pi_{\theta_k} \|_\infty \leq 2 \| Q^{\text{soft}}_{r_{\theta_k}, \pi_k} - Q^{\text{soft}}_{r_{\theta_k}, \pi_{\theta_k}} \|_\infty
\tag{62}
$$

where $\| \log \pi_{k+1} - \log \pi_{\theta_k} \|_\infty = \max_{s \in \mathcal{S}, a \in \mathcal{A}} | \log \pi_{k+1}(a|s) - \log \pi_{\theta_k}(a|s)|$ and $\| Q^{\text{soft}}_{r_{\theta_k}, \pi_k} - Q^{\text{soft}}_{r_{\theta_k}, \pi_{\theta_k}} \|_\infty = \max_{s \in \mathcal{S}, a \in \mathcal{A}} |Q^{\text{soft}}_{r_{\theta_k}, \pi_k}(s,a) - Q^{\text{soft}}_{r_{\theta_k}, \pi_{\theta_k}}(s,a)|$.

Based on the inequality (62), we analyze $\| Q^{\text{soft}}_{r_{\theta_k}, \pi_k} - Q^{\text{soft}}_{r_{\theta_k}, \pi_{\theta_k}} \|_\infty$ to show the convergence of the policy estimates. It leads to the following analysis:

$$
\begin{aligned}
& \| Q^{\text{soft}}_{r_{\theta_k}, \pi_k} - Q^{\text{soft}}_{r_{\theta_k}, \pi_{\theta_k}} \|_\infty \\
& = \| Q^{\text{soft}}_{r_{\theta_k}, \pi_k} - Q^{\text{soft}}_{r_{\theta_k}, \pi_{\theta_k}} + Q^{\text{soft}}_{r_{\theta_{k-1}}, \pi_{\theta_{k-1}}} - Q^{\text{soft}}_{r_{\theta_{k-1}}, \pi_{\theta_{k-1}}} + Q^{\text{soft}}_{r_{\theta_{k-1}}, \pi_k} - Q^{\text{soft}}_{r_{\theta_{k-1}}, \pi_k} \|_\infty \\
& \leq \| Q^{\text{soft}}_{r_{\theta_k}, \pi_{\theta_k}} - Q^{\text{soft}}_{r_{\theta_{k-1}}, \pi_{\theta_{k-1}}} \|_\infty + \| Q^{\text{soft}}_{r_{\theta_{k-1}}, \pi_k} - Q^{\text{soft}}_{r_{\theta_{k-1}}, \pi_{\theta_{k-1}}} \|_\infty + \| Q^{\text{soft}}_{r_{\theta_k}, \pi_k} - Q^{\text{soft}}_{r_{\theta_{k-1}}, \pi_k} \|_\infty \\
& \stackrel{(i)}{\leq} L_q \|\theta_k - \theta_{k-1}\| + \| Q^{\text{soft}}_{r_{\theta_{k-1}}, \pi_k} - Q^{\text{soft}}_{r_{\theta_{k-1}}, \pi_{\theta_{k-1}}} \|_\infty + \| Q^{\text{soft}}_{r_{\theta_k}, \pi_k} - Q^{\text{soft}}_{r_{\theta_{k-1}}, \pi_k} \|_\infty \\
& \stackrel{(ii)}{\leq} \| Q^{\text{soft}}_{r_{\theta_{k-1}}, \pi_k} - Q^{\text{soft}}_{r_{\theta_{k-1}}, \pi_{\theta_{k-1}}} \|_\infty + 2 L_q \|\theta_k - \theta_{k-1}\|
\end{aligned}
\tag{63}
$$

where (i) is from (39a) in Lemma A.1; (ii) follows (39a). Based on (63), we further analyze the two terms in (63) as below.

Recall we have the soft Bellman operator expressed as below:

$$\mathcal{T}_\theta(Q)(s,a) = r(s,a;\theta) + \gamma\mathbb{E}_{s'\sim P(\cdot|s',a')}\left[\log\left(\sum_{a'}\pi^0(a'|s')\exp\left(Q(s',a')\right)\right)\right] \quad (64)$$

According to the soft Bellman operator, it holds that

$$
\begin{aligned}
Q^{\text{soft}}_{r_{\theta_k},\pi_{k+1}}(s,a) &= r(s,a;\theta_k) + \gamma\mathbb{E}_{s'\sim\mathcal{P}(\cdot|s,a)}[V^{\text{soft}}_{r_{\theta_k},\pi_{k+1}}(s')] \\
&= r(s,a;\theta_k) + \gamma\mathbb{E}_{s'\sim\mathcal{P}(\cdot|s,a),a'\sim\pi_{k+1}(\cdot|s')}[-\frac{\log\pi_{k+1}(a'|s')}{\log\pi_0(a'|s')} + Q^{\text{soft}}_{r_{\theta_k},\pi_{k+1}}(s',a')] \\
&\overset{(i)}{\geq} r(s,a;\theta_k) + \gamma\mathbb{E}_{s'\sim\mathcal{P}(\cdot|s,a),a'\sim\pi_{k+1}(\cdot|s')}[-\frac{\log\pi_{k+1}(a'|s')}{\log\pi_0(a'|s')} + Q^{\text{soft}}_{r_{\theta_k},\pi_k}(s',a')] \\
&\overset{(ii)}{=} r(s,a;\theta_k) + \gamma\mathbb{E}_{s'\sim\mathcal{P}(\cdot|s,a)}\left[\log\left(\sum_{a'}\pi^0(a'|s')\exp\left(Q^{\text{soft}}_{r_{\theta_k},\pi_k}(s',a')\right)\right)\right] \\
&\overset{(iii)}{=} \mathcal{T}_{\theta_k}(Q^{\text{soft}}_{r_{\theta_k},\pi_k})(s,a)
\end{aligned}
\quad (65)
$$

where (i) follows the policy improvement result(ii) follows the definition $\pi_{k+1}(a|s) :=$ $\frac{\pi^0(a|s)\exp\left(Q^{\text{soft}}_{r_{\theta_k},\pi_k}(s,a)\right)}{\sum_{\tilde{a}}\pi^0(\tilde{a}|s)\exp\left(Q^{\text{soft}}_{r_{\theta_k},\pi_k}(s,\tilde{a})\right)}$ (iii) follows the definition of the soft Bellman operator in (64).

For any $s\in\mathcal{S}$ and $a\in\mathcal{A}$, it holds that

$$0 \overset{(i)}{\leq} Q^{\text{soft}}_{r_{\theta_k},\pi_{\theta_k}}(s,a) - Q^{\text{soft}}_{r_{\theta_k},\pi_{k+1}}(s,a) \overset{(ii)}{\leq} Q^{\text{soft}}_{r_{\theta_k},\pi_{\theta_k}}(s,a) - \mathcal{T}_{\theta_k}(Q^{\text{soft}}_{r_{\theta_k},\pi_k})(s,a) \quad (66)$$

where (i) is due to the fact that $\pi_{\theta_k}$ is the optimal policy under reward parameter $\theta_k$; (ii) is from (65). Hence, it further leads to

$$
\begin{aligned}
\|Q^{\text{soft}}_{r_{\theta_k},\pi_{\theta_k}} - Q^{\text{soft}}_{r_{\theta_k},\pi_{k+1}}\|_\infty &\overset{(i)}{\leq} \|Q^{\text{soft}}_{r_{\theta_k},\pi_{\theta_k}} - \mathcal{T}_{\theta_k}(Q^{\text{soft}}_{r_{\theta_k},\pi_k})\|_\infty \\
&\overset{(ii)}{=} \|\mathcal{T}_{\theta_k}(Q^{\text{soft}}_{r_{\theta_k},\pi_{\theta_k}}) - \mathcal{T}_{\theta_k}(Q^{\text{soft}}_{r_{\theta_k},\pi_k})\|_\infty \\
&\overset{(iii)}{\leq} \gamma\|Q^{\text{soft}}_{r_{\theta_k},\pi_{\theta_k}} - Q^{\text{soft}}_{r_{\theta_k},\pi_k}\|_\infty
\end{aligned}
\quad (67)
$$

where (i) is from (66); (ii) is from the fixed-point property in (83); (iii) is from the contraction property in (82). Therefore, we have the following result:

$$
\begin{aligned}
&\|Q^{\text{soft}}_{r_{\theta_k},\pi_k} - Q^{\text{soft}}_{r_{\theta_k},\pi_{\theta_k}}\|_\infty \\
&\overset{(i)}{\leq} \|Q^{\text{soft}}_{r_{\theta_{k-1}},\pi_k} - Q^{\text{soft}}_{r_{\theta_{k-1}},\pi_{\theta_{k-1}}}\|_\infty + 2L_q\|\theta_k - \theta_{k-1}\| \\
&\overset{(ii)}{\leq} \gamma\|Q^{\text{soft}}_{r_{\theta_{k-1}},\pi_{k-1}} - Q^{\text{soft}}_{r_{\theta_{k-1}},\pi_{\theta_{k-1}}}\|_\infty + 2L_q\|\theta_k - \theta_{k-1}\|
\end{aligned}
\quad (68)
$$

where (i) is from (63); (ii) is from (67).

To show the convergence of the soft Q-function based on (68), we further analyze the error between the reward parameters $\theta_k$ and $\theta_{k-1}$. Recall in Alg.1, the updates in reward parameters (19):

$$\theta_k = \theta_{k-1} + \alpha g_{k-1}$$

where we denote $\tau = \{(s_t,a_t)\}_{t=0}^\infty$, $h(\theta,\tau) := \sum_{t\geq 0}\gamma^t\nabla_\theta r(s_t,a_t;\theta)$ and $g_{k-1}$ is the stochastic gradient estimator at iteration $k-1$. Here, $\tau^E_{k-1}$ denotes the trajectory sampled from the expert's dataset $D$ at iteration $k-1$ and $\tau^A_{k-1}$ denotes the trajectory sampled from the agent's policy $\pi_k$ at

time $k-1, \tau_w, \tau_i$ denote the trajectory sampled from the preference dataset. Then according to the inequality (38a) in Assumption 2, we could show that

$$\|g_{k-1}\| \le \|h(\theta_{k-1}, \tau_{k-1}^E) - h(\theta_{k-1}, \tau_{k-1}^A)\| + \|h(\theta_{k-1}, \tau_{k-1}^W) - h(\theta_{k-1}, \tau_{k-1}^L)\|$$

$$\le \frac{2L_r}{1-\gamma} + \frac{2L_r}{1-\gamma} = 4L_q \tag{69}$$

where the last equality follows the fact that we have defined the constant $L_q := \frac{L_r}{1-\gamma}$. Then we could further show that

$$\|Q_{r_{\theta_k},\pi_k}^{\text{soft}} - Q_{r_{\theta_k},\pi_{\theta_k}}^{\text{soft}}\|_\infty$$

$$\overset{(i)}{\le} \gamma\|Q_{r_{\theta_{k-1}},\pi_{k-1}}^{\text{soft}} - Q_{r_{\theta_{k-1}},\pi_{\theta_{k-1}}}^{\text{soft}}\|_\infty + 4L_q\|\theta_k - \theta_{k-1}\|$$

$$\overset{(ii)}{=} \gamma\|Q_{r_{\theta_{k-1}},\pi_{k-1}}^{\text{soft}} - Q_{r_{\theta_{k-1}},\pi_{\theta_{k-1}}}^{\text{soft}}\|_\infty + 4\alpha L_q\|g_{k-1}\|$$

$$\overset{(iii)}{\le} \gamma\|Q_{r_{\theta_{k-1}},\pi_{k-1}}^{\text{soft}} - Q_{r_{\theta_{k-1}},\pi_{\theta_{k-1}}}^{\text{soft}}\|_\infty + 8\alpha L_q^2 \tag{70}$$

where (i) is from (68); (ii) follows the reward update scheme; (iii) is from (69).

Summing the inequality (70) from $k=1$ to $k=K$, it holds that

$$\sum_{k=1}^{K}\|Q_{r_{\theta_k},\pi_k}^{\text{soft}} - Q_{r_{\theta_k},\pi_{\theta_k}}^{\text{soft}}\|_\infty \le \gamma \sum_{k=0}^{K-1}\|Q_{r_{\theta_k},\pi_k}^{\text{soft}} - Q_{r_{\theta_k},\pi_{\theta_k}}^{\text{soft}}\|_\infty + 8\alpha K L_q^2 \tag{71}$$

Rearranging the inequality (71) and divided (71) by $K$ on both sides, it holds that

$$\frac{1-\gamma}{K}\sum_{k=1}^{K}\|Q_{r_{\theta_k},\pi_k}^{\text{soft}} - Q_{r_{\theta_k},\pi_{\theta_k}}^{\text{soft}}\|_\infty \le \frac{\gamma}{K}\left(\|Q_{r_{\theta_0},\pi_0}^{\text{soft}} - Q_{r_{\theta_0},\pi_{\theta_0}}^{\text{soft}}\|_\infty - \|Q_{r_{\theta_K},\pi_K}^{\text{soft}} - Q_{r_{\theta_K},\pi_{\theta_K}}^{\text{soft}}\|_\infty\right) + 8\alpha L_q^2 \tag{72}$$

Dividing the constant $1-\gamma$ on both sides of (72), it holds that

$$\frac{1}{K}\sum_{k=1}^{K}\|Q_{r_{\theta_k},\pi_k}^{\text{soft}} - Q_{r_{\theta_k},\pi_{\theta_k}}^{\text{soft}}\|_\infty \le \frac{\gamma C_0}{K(1-\gamma)} + \frac{8L_q^2}{1-\gamma}\alpha$$

where we denote $C_0 := \|Q_{r_{\theta_0},\pi_0}^{\text{soft}} - Q_{r_{\theta_0},\pi_{\theta_0}}^{\text{soft}}\|_\infty$. We could also write the inequality above as

$$\frac{1}{K}\sum_{k=0}^{K-1}\|Q_{r_{\theta_k},\pi_k}^{\text{soft}} - Q_{r_{\theta_k},\pi_{\theta_k}}^{\text{soft}}\|_\infty$$

$$\le \frac{\gamma C_0}{K(1-\gamma)} + \frac{C_0}{K} - \frac{\|Q_{r_{\theta_K},\pi_K}^{\text{soft}} - Q_{r_{\theta_K},\pi_{\theta_K}}^{\text{soft}}\|_\infty}{K} + \frac{8L_q^2}{1-\gamma}\alpha$$

$$\le \frac{C_0}{K(1-\gamma)} + \frac{8L_q^2}{1-\gamma}\alpha.$$

Recall the stepsize is defined as $\alpha = \frac{\alpha_0}{K^\sigma}$ where $\sigma > 0$. Then we have the following result:

$$\frac{1}{K}\sum_{k=0}^{K-1}\|Q_{r_{\theta_k},\pi_k}^{\text{soft}} - Q_{r_{\theta_k},\pi_{\theta_k}}^{\text{soft}}\|_\infty = \mathcal{O}(K^{-1}) + \mathcal{O}(K^{-\sigma}). \tag{73}$$

With the inequality (62), it follows that

$$\frac{1}{K}\sum_{k=0}^{K-1}\|\log\pi_{k+1} - \log\pi_{\theta_k}\|_\infty \le \frac{2}{K}\sum_{k=0}^{K-1}\|Q_{r_{\theta_k},\pi_k}^{\text{soft}} - Q_{r_{\theta_k},\pi_{\theta_k}}^{\text{soft}}\|_\infty = \mathcal{O}(K^{-1}) + \mathcal{O}(K^{-\sigma}).$$

Therefore, we complete the proof of (20a) in Theorem 4.1.

### A.5.2 PROOF OF (20B)

In this part, we prove the convergence of reward parameters $\{\theta_k\}_{k \geq 0}$.

We have the following result of the objective function $L(\theta)$:

$$
\begin{aligned}
L(\theta_{k+1}) &\overset{(i)}{\geq} L(\theta_k) + \langle \nabla L(\theta_k), \theta_{k+1} - \theta_k \rangle - \frac{L_c}{2}\|\theta_{k+1} - \theta_k\|^2 \\
&\overset{(ii)}{=} L(\theta_k) + \alpha \langle \nabla L(\theta_k), g_k \rangle - \frac{L_c \alpha^2}{2}\|g_k\|^2 \\
&= L(\theta_k) + \alpha \langle \nabla L(\theta_k), g_k - \nabla L(\theta_k) \rangle + \alpha\|\nabla L(\theta_k)\|^2 - \frac{L_c \alpha^2}{2}\|g_k\|^2 \\
&\overset{(iii)}{\geq} L(\theta_k) + \alpha \langle \nabla L(\theta_k), g_k - \nabla L(\theta_k) \rangle + \alpha\|\nabla L(\theta_k)\|^2 - 8L_c L_q^2 \alpha^2
\end{aligned}
\tag{74}
$$

where (i) is from the Lipschitz smooth property in (39b) of Lemma A.1; (ii) follows the update scheme (19); (iii) is from constant bound in (69). Taking an expectation over the both sides of (74), it holds that

$$
\mathbb{E}\left[L(\theta_{k+1})\right]
$$

$$
\geq \mathbb{E}\left[L(\theta_k)\right] + \alpha\mathbb{E}\left[\langle \nabla L(\theta_k), g_k - \nabla L(\theta_k) \rangle\right] + \alpha\mathbb{E}\left[\|\nabla L(\theta_k)\|^2\right] - 8L_c L_q^2 \alpha^2
$$

$$
= \mathbb{E}\left[L(\theta_k)\right] + \alpha\mathbb{E}\left[\langle \nabla L(\theta_k), \mathbb{E}[g_k - \nabla L(\theta_k)|\theta_k] \rangle\right] + \alpha\mathbb{E}\left[\|\nabla L(\theta_k)\|^2\right] - 8L_c L_q^2 \alpha^2
$$

$$
= \mathbb{E}\left[L(\theta_k)\right] + \alpha\mathbb{E}\left[\left\langle \nabla L(\theta_k), \mathbb{E}_{\tau \sim \pi_{\theta_k}}\left[\sum_{t \geq 0} \gamma^t \nabla_\theta r(s_t, a_t; \theta_k)\right] - \mathbb{E}_{\tau \sim \pi_{k+1}}\left[\sum_{t \geq 0} \gamma^t \nabla_\theta r(s_t, a_t; \theta_k)\right]\right.\right.
$$

$$
\left.\left. + \mathbb{E}_{(\tau_l \prec \tau_w) \sim \pi^P}\left[\sum_{t \geq 0}(1 - \sigma(\gamma^t r(s_t^w, a_t^w; \theta_k) - \gamma^t r(s_t^l, a_t^l; \theta_k)))(\gamma^t \nabla_\theta r(s_t^w, a_t^w; \theta_k) - \gamma^t \nabla_\theta r(s_t^l, a_t^l; \theta_k))\right]\right\rangle\right]
$$

$$
+ \alpha\mathbb{E}\left[\|\nabla L(\theta_k)\|^2\right] - 8L_c L_q^2 \alpha^2
$$

$$
\overset{(i)}{\geq} \mathbb{E}\left[L(\theta_k)\right] - 4\alpha L_q \underbrace{\mathbb{E}\left[\left\|\mathbb{E}_{\tau \sim \pi_{\theta_k}}\left[\sum_{t \geq 0} \gamma^t \nabla_\theta r(s_t, a_t; \theta_k)\right] - \mathbb{E}_{\tau \sim \pi_{k+1}}\left[\sum_{t \geq 0} \gamma^t \nabla_\theta r(s_t, a_t; \theta_k)\right]\right\|\right]}_{\text{term A}}
$$

$$
+ \alpha\mathbb{E}\left[\|\nabla L(\theta_k)\|^2\right] - 8L_c L_q^2 \alpha^2
\tag{75}
$$

(i) is due to the fact that $\|\nabla L(\theta)\| \leq 4L_q$ and $\mathbb{E}[g_{k,2} - \nabla_\theta L_2(\theta_k)|\theta_k] = 0$.

Then we further analyze the term A as below:

$$
\mathbb{E}\left[\left\|\mathbb{E}_{\tau\sim\pi_{\theta_k}}\left[\sum_{t\geq 0}\gamma^t\nabla_\theta r(s_t,a_t;\theta_k)\right]-\mathbb{E}_{\tau\sim\pi_{k+1}}\left[\sum_{t\geq 0}\gamma^t\nabla_\theta r(s_t,a_t;\theta_k)\right]\right\|\right]
$$

$$
\overset{(i)}{=}\mathbb{E}\left[\left\|\frac{1}{1-\gamma}\mathbb{E}_{(s,a)\sim d(\cdot,\cdot;\pi_{\theta_k})}\left[\nabla_\theta r(s,a;\theta_k)\right]-\frac{1}{1-\gamma}\mathbb{E}_{(s,a)\sim d(\cdot,\cdot;\pi_{k+1})}\left[\nabla_\theta r(s,a;\theta_k)\right]\right\|\right]
$$

$$
\overset{(ii)}{\leq}\frac{2}{1-\gamma}\cdot\max_{s\in\mathcal{S},a\in\mathcal{A}}\|\nabla_\theta r(s,a;\theta_k)\|\cdot\mathbb{E}\left[\|d(\cdot,\cdot;\pi_{\theta_k})-d(\cdot,\cdot;\pi_{k+1})\|_{TV}\right]
$$

$$
\overset{(iii)}{\leq}\frac{2L_r}{1-\gamma}\mathbb{E}\left[\|d(\cdot,\cdot;\pi_{\theta_k})-d(\cdot,\cdot;\pi_{k+1})\|_{TV}\right]
$$

$$
\overset{(iv)}{\leq}2L_qC_d\mathbb{E}\left[\|\log\frac{\pi^0(a|s)\exp Q^{\mathrm{soft}}_{r_{\theta_k},\pi_{\theta_k}}(s,a)}{\sum_{\tilde a}\pi^0(\tilde a|s)\exp Q^{\mathrm{soft}}_{r_{\theta_k},\pi_{\theta_k}}(s,\tilde a)}-\log\frac{\pi^0(a|s)\exp Q^{\mathrm{soft}}_{r_{\theta_k},\pi_{k+1}}(s,a)}{\sum_{\tilde a}\pi^0(\tilde a|s)\exp Q^{\mathrm{soft}}_{r_{\theta_k},\pi_{k+1}}(s,\tilde a)}\|\right]
$$

$$
\overset{(v)}{\leq}2L_qC_d\mathbb{E}\left[\|Q^{\mathrm{soft}}_{r_{\theta_k},\pi_{\theta_k}}-Q^{\mathrm{soft}}_{r_{\theta_k},\pi_k}\|+\|\log\sum_a\pi^0(\tilde a|s)\exp Q^{\mathrm{soft}}_{r_{\theta_k},\pi_{\theta_k}}(s,\tilde a)-\log\sum_a\pi^0(\tilde a|s)\exp Q^{\mathrm{soft}}_{r_{\theta_k},\pi_{k+1}}(s,\tilde a)\|\right]
$$

$$
\overset{(vi)}{\leq}2L_qC_d\sqrt{|\mathcal{S}|\cdot|\mathcal{A}|}\mathbb{E}\left[\|Q^{\mathrm{soft}}_{r_{\theta_k},\pi_{\theta_k}}-Q^{\mathrm{soft}}_{r_{\theta_k},\pi_k}\|_\infty+\|Q^{\mathrm{soft}}_{r_{\theta_k},\pi_{\theta_k}}-Q^{\mathrm{soft}}_{r_{\theta_k},\pi_k}\|_\infty\right]
$$

$$
=4L_qC_d\sqrt{|\mathcal{S}|\cdot|\mathcal{A}|}\mathbb{E}\left[\|Q^{\mathrm{soft}}_{r_{\theta_k},\pi_{\theta_k}}-Q^{\mathrm{soft}}_{r_{\theta_k},\pi_k}\|_\infty\right]\tag{76}
$$

where (i) follows the definition $d(s,a;\pi)=(1-\gamma)\pi(a|s)\sum_{t\geq 0}\gamma^t\mathcal{P}^\pi(s_t=s|s_0\sim\eta)$; (ii) is due to distribution mismatch between two visitation measures; (iii) follows the inequality (38a) in Assumption 2; the inequality (iv) follows Lemma A.2 and the fact that $\pi_{\theta_k}(\cdot|s)\propto\pi^0(\cdot|s)\exp\left(Q^{\mathrm{soft}}_{r_{\theta_k},\pi_{\theta_k}}(s,\cdot)\right)$, $\pi_{k+1}(\cdot|s)\propto\pi^0(\cdot|s)\exp\left(Q^{\mathrm{soft}}_{r_{\theta_k},\pi_k}(s,\cdot)\right)$ and the constant $L_q:=\frac{L_r}{1-\gamma}$; (v) follows the (60);(vi) follows the conversion between Frobenius norm and infinity norm.

Through plugging the inequality (76) into (75), it leads to

$$
\mathbb{E}\left[L(\theta_{k+1})\right]
$$

$$
\geq\mathbb{E}\left[L(\theta_k)\right]-2\alpha L_q\mathbb{E}\left[\left\|\mathbb{E}_{\tau\sim\pi_{\theta_k}}\left[\sum_{t\geq 0}\gamma^t\nabla_\theta r(s_t,a_t;\theta_k)\right]-\mathbb{E}_{\tau\sim\pi_{k+1}}\left[\sum_{t\geq 0}\gamma^t\nabla_\theta r(s_t,a_t;\theta_k)\right]\right\|\right]
$$

$$
+\alpha\mathbb{E}\left[\|\nabla L(\theta_k)\|^2\right]-8L_cL_q^2\alpha^2
$$

$$
\overset{(i)}{\geq}\mathbb{E}\left[L(\theta_k)\right]-8\alpha C_dL_q^2\sqrt{|\mathcal{S}|\cdot|\mathcal{A}|}\mathbb{E}\left[\|Q^{\mathrm{soft}}_{r_{\theta_k},\pi_{\theta_k}}-Q^{\mathrm{soft}}_{r_{\theta_k},\pi_k}\|_\infty\right]+\alpha\mathbb{E}\left[\|\nabla L(\theta_k)\|^2\right]-8L_cL_q^2\alpha^2
$$

where (i) follows the inequality (76).

Rearranging the inequality above and denote $C_1:=8C_dL_q^2\sqrt{|\mathcal{S}|\cdot|\mathcal{A}|}$, it holds that

$$
\alpha\mathbb{E}\left[\|\nabla L(\theta_k)\|^2\right]\leq 8L_cL_q^2\alpha^2+\alpha C_1\mathbb{E}\left[\|Q^{\mathrm{soft}}_{r_{\theta_k},\pi_{\theta_k}}-Q^{\mathrm{soft}}_{r_{\theta_k},\pi_k}\|_\infty\right]+\mathbb{E}\left[L(\theta_{k+1})-L(\theta_k)\right]
$$

Summing the inequality above from $k=0$ to $K-1$ and dividing both sides by $\alpha K$, it holds that

$$
\frac{1}{K}\sum_{k=0}^{K-1}\mathbb{E}\left[\|\nabla L(\theta_k)\|^2\right]\leq 8L_cL_q^2\alpha+\frac{C_1}{K}\sum_{k=0}^{K-1}\mathbb{E}\left[\|Q^{\mathrm{soft}}_{r_{\theta_k},\pi_{\theta_k}}-Q^{\mathrm{soft}}_{r_{\theta_k},\pi_k}\|_\infty\right]+\mathbb{E}\left[\frac{L(\theta_K)-L(\theta_0)}{K\alpha}\right]\tag{77}
$$

Note that the log-likelihood function $L(\theta_K)$ is negative and $L(\theta_0)$ is a bounded constant. Then we could plug (73) into (77), it holds that

$$
\frac{1}{K}\sum_{K=0}^{K-1}\mathbb{E}\left[\|\nabla L(\theta_K)\|^2\right]=\mathcal{O}(K^{-\sigma})+\mathcal{O}(K^{-1})+\mathcal{O}(K^{-1+\sigma})\tag{78}
$$

which completes the proof for the inequality (20b).

A.6   AUXILIARY LEMMAS

**Lemma A.2** *((Xu et al., 2020, Lemma 3)) Consider the initialization distribution $\eta(\cdot)$ and transition kernel $\mathcal{P}(\cdot|s, a)$. Under $\eta(\cdot)$ and $\mathcal{P}(\cdot|s, a)$, denote $d_w(\cdot, \cdot)$ as the state-action visitation distribution of MDP with the Boltzman policy parameterized by parameter $w$. Suppose Assumption 1 holds, for all policy parameter $w$ and $w'$, we have*

$$\|d_w(\cdot, \cdot) - d_{w'}(\cdot, \cdot)\|_{TV} \leq C_d \|w - w'\| \tag{79}$$

*where $C_d$ is a positive constant.*

Next, to facilitate analysis for KL-regularized MDPs, we introduce a soft Bellman optimality operator $\mathcal{T} : \mathbb{R}^{|\mathcal{S}| \times |\mathcal{A}|} \to \mathbb{R}^{|\mathcal{S}| \times |\mathcal{A}|}$ as follows:

$$\mathcal{T}(Q)(s, a) := r(s, a) + \gamma \mathbb{E}_{s' \sim \mathcal{P}(\cdot|s,a)} \left[ \max_{\pi(\cdot|s)} \mathbb{E}_{a' \sim \pi(\cdot|s')} \left[ Q(s', a') - \frac{\log \pi(a'|s')}{\log \pi^0(a'|s')} \right] \right]. \tag{80}$$

In the following lemma, the properties of KL-regularized MDPs are characterized.

**Lemma A.3** *(The operator $\mathcal{T}$ as defined in (80) satisfies the properties below:*

- *$\mathcal{T}$ has the following closed-form expression:*

$$\mathcal{T}(Q)(s, a) = r(s, a) + \gamma \mathbb{E}_{s' \sim \mathcal{P}(\cdot|s,a)} \left[ \log \left( \sum_{a'} \pi^0(a'|s') \exp\left( Q(s', a') \right) \right) \right]. \tag{81}$$

- *$\mathcal{T}$ is a $\gamma$-contraction in the $\ell_\infty$ norm, namely, for any $Q_1, Q_2 \in \mathbb{R}^{|\mathcal{S}| \times |\mathcal{A}|}$, it holds that*

$$\|\mathcal{T}(Q_1) - \mathcal{T}(Q_2)\|_\infty \leq \gamma \|Q_1 - Q_2\|_\infty. \tag{82}$$

- *Under a given reward function $r(\cdot, \cdot)$, the corresponding optimal soft Q-function $Q_{r,\pi^*}^{soft}$ is a unique fixed point of the operator $\mathcal{T}$, namely,*

$$\mathcal{T}(Q_{r,\pi^*}^{soft}) = Q_{r,\pi^*}^{soft} \tag{83}$$

We refine its analysis as below.

We first show that

$$\mathbb{E}_{a \sim \pi(\cdot|s)} \left[ Q(s, a) - \frac{\log \pi(a|s)}{\log \pi^0(a|s)} \right] = \sum_a \pi(a|s) \log \left( \frac{\pi^0(a|s) \exp(Q(s, a))}{\pi(a|s)} \right) \overset{(i)}{\leq} \log \left( \sum_a \pi^0(a|s) \exp\left( Q(s, a) \right) \right) \tag{84}$$

where (i) is from Jensen's inequality. Moreover, the equality between both sides of (i) holds when the policy $\pi$ has the expression $\pi(\cdot|s) \propto \pi^0(a|s) \exp(Q(s, \cdot))$. Therefore, through applying the inequality (84) to (80), it obtains that

$$\mathcal{T}(Q)(s, a) = r(s, a) + \gamma \mathbb{E}_{s' \sim \mathcal{P}(\cdot|s,a)} \left[ \log \left( \sum_{a'} \pi^0(a|s) \exp\left( Q(s', a') \right) \right) \right], \tag{85}$$

which proves the equality (81).

We define $\|Q_1 - Q_2\|_\infty := \max_{s \in \mathcal{S}, a \in \mathcal{A}} |Q_1(s, a) - Q_2(s, a)|$ and $\epsilon = \|Q_1 - Q_2\|_\infty$. Then for any $s \in \mathcal{S}$ and $a \in \mathcal{A}$, it follows that

$$\log \left( \sum_a \pi^0(a|s) \exp\left( Q_1(s, a) \right) \right) \leq \log \left( \sum_a \pi^0(a|s) \exp\left( Q_2(s, a) + \epsilon \right) \right)$$

$$= \log \left( \exp(\epsilon) \sum_a \pi^0(a|s) \exp\left( Q_2(s, a) \right) \right)$$

$$= \epsilon + \log \left( \sum_a \pi^0(a|s) \exp\left( Q_2(s, a) \right) \right)$$

Similarly, it is easy to obtain that $\log \left( \sum_a \pi^0(a|s) \exp \left( Q_1(s,a) \right) \right) \geq -\epsilon + \log \left( \sum_a \pi^0(a|s) \exp \left( Q_2(s,a) \right) \right)$. Hence, it leads to the contraction property that

$$\|\mathcal{T}(Q_1) - \mathcal{T}(Q_2)\|_\infty \leq \gamma\epsilon = \gamma\|Q_1 - Q_2\|_\infty \tag{86}$$

which proves the contraction property (82). Moreover, we have

$$\mathcal{T}(Q_{r,\pi^*}^{\mathrm{soft}})(s,a) \stackrel{(i)}{=} r(s,a) + \gamma\mathbb{E}_{s'\sim\mathcal{P}(\cdot|s,a)}\left[ \log \left( \sum_{a'} \pi^0(a'|s') \exp \left( Q_{r,\pi^*}^{\mathrm{soft}}(s',a') \right) \right) \right] \stackrel{(ii)}{=} Q_{r,\pi^*}^{\mathrm{soft}}(s,a) \tag{87}$$

where (i) follows the equality (85). Based on the definition of the soft Q-function $Q_{r,\pi^*}^{\mathrm{soft}}$, we have

$$Q_{r,\pi^*}^{\mathrm{soft}}(s,a) = r(s,a) + \gamma\mathbb{E}_{s'\sim\mathcal{P}(\cdot|s,a)}\left[ \mathbb{E}_{a'\sim\pi^*(\cdot|s')}[-\frac{\log \pi^*(a'|s')}{\log \pi^0(a'|s')} + Q_{r,\pi^*}^{\mathrm{soft}}(s',a')]\right]. \tag{88}$$

We prove the equality (ii) in (87) through combining (88) and the fact that the optimal soft policy has the closed form $\pi^*(\cdot|s) \propto \pi^0(\cdot|s') \exp \left( Q_{r,\pi^*}^{\mathrm{soft}}(s,\cdot) \right)$. Suppose two different fixed points of the soft Bellman operator exist, then it contradicts with the contraction property in (86).

Hence, we proved the uniqueness of the optimal soft Q-function $Q_{r,\pi^*}^{\mathrm{soft}}$. Moreover, the optimal soft Q-function $Q_{r,\pi^*}^{\mathrm{soft}}$ is a fixed point to the soft Bellman operator $\mathcal{T}$ in (83).

