# OpenReview forum: "Joint Reward and Policy Learning with Demonstrations and Human Feedback Improves Alignment"
_ICLR.cc/2025/Conference — ICLR 2025 Spotlight_

### Official Review · Reviewer_1nbf · 2024-10-16

**Soundness:** 4
**Presentation:** 4
**Contribution:** 4
**Rating:** 10
**Confidence:** 4

**Summary:**

This paper proposes a new framework that utilizes both demonstration data and preference data for better alignment. This idea is novel and interesting. Moreover, the paper also provides a general theoretical bi-level formulation that not only induces the proposed AIHF, but also can reduce to some major RLHF and IRL methods as special cases. The paper proposes an efficient single-loop algorithm to solve the bi-level optimization problem and theoretically guarantee the finite-time convergence of the proposed algorithm. Extensive empirical evaluations are provided to validate the effectiveness of the proposed method.

**Strengths:**

This paper proposes a novel and effective integration of IRL and RLHF to improvement alignment. I am actually very happy with this paper due to this novel integration and the associated theoretical framework. Moreover, I think that this paper opens a door for future research on the integration of IRL and RLHF for better alignment. The strengths of this paper include:
1. Novel and interesting idea of the integration of IRL and RLHF for better alignment.
2. A general bi-level formulation of this integration which can also reduce to some major IRL and RLHF methods as special cases.
3. Excellent presentation where the authors explicitly deliver their ideas. More importantly, the authors provide insights to help readers better understand why the proposed framework can lead to better alignment. These insights in Section 3.4 are very helpful for readers to get an initial understanding of the advantages of the proposed framework.
4. Solid theoretical guarantee for the proposed algorithm.

In general, I think that this paper can contribute to the RLHF and IRL community.

**Weaknesses:**

There is no obvious weakness of this paper. Please see questions.

**Questions:**

1. In the introduction (lines 63-64), it is said that "a joint approach to learning reward and policy models may improve alignment at
the expense of potentially significant additional computational effort". At that time, I expected that this method would require additional demonstration data, compared to the standard two-stage RLHF method. However, Figure 1 shows that the demonstration data in AIHF is the demonstration data used for SFT in the standard RLHF, so that there is no additional demonstration data needed? Then I am not sure why the proposed method may potentially lead to additional computation. Of course, the computation is higher compared to RLHF using preference data only. However, RLHF also needs demonstration data to first compute SFT policy. If we compare "AIHF" and "SFT+RLHF", intuitively "SFT+RLHF" will be more computationally expensive because it uses the same amount data as AIHF and it solves two separate optimization problems. The counterpart of AIHF is not RLHF but RLHF+SFT, right? So that we need to compare AIHF and RLHF+SFT.

2. In Section 3.4, it is shown that AIHF policy is somehow a weighted average of IRL policy from demonstrations and RLHF policy from preferences, therefore the AIHF policy reduces variance. I agree that this average can reduce variance. Suppose the demonstration needed for SFT is the same demonstration data in AIHF, the standard RLHF has a KL regularization $D_{KL}(\pi||\pi_{SFT})$ (which relates the learned policy $\pi$ to the policy $\pi_{SFT}$ learned from demonstration data). If we linearize the KL regularization, this may also lead to a (weight) average of $\pi_{SFT}$ (demonstration) and the RLHF policy learned from preference?

---

> ### Author Response · Authors · 2024-11-23
>
> We thank the reviewer for your positive comments and recognizing the importance of this work. Below, we address the reviewer's comments in a point-by-point manner.
>
> > 1. In the introduction (lines 63-64), it is said that "a joint approach to learning reward and policy models may improve alignment at the expense of potentially significant additional computational effort". At that time, I expected that this method would require additional demonstration data, compared to the standard two-stage RLHF method. However, Figure 1 shows that the demonstration data in AIHF is the demonstration data used for SFT in the standard RLHF, so that there is no additional demonstration data needed? Then I am not sure why the proposed method may potentially lead to additional computation. Of course, the computation is higher compared to RLHF using preference data only. However, RLHF also needs demonstration data to first compute SFT policy. If we compare "AIHF" and "SFT+RLHF", intuitively "SFT+RLHF" will be more computationally expensive because it uses the same amount data as AIHF and it solves two separate optimization problems. The counterpart of AIHF is not RLHF but RLHF+SFT, right? So that we need to compare AIHF and RLHF+SFT.
>
> **Response**: We appreciate the reviewer's insightful comment.  We'd like to clarify that indeed there is no additional demonstration data needed for AIHF comparing to RLHF+SFT. Throughout the paper, the comparison is fair in the sense that data used are the same comparing RLHF and AIHF. In AIHF, the demonstration data we use is actually the SFT dataset which is used in the RLHF pipeline to train one SFT model. Moreover, we also notice that many public models are trained using more demonstration data compared with the preference data (as we show in the general response to all reviewers). One potential reason why many public models are trained with more demonstrations is that demonstrations can sometimes be collected from existing datasets or created by experts in a more efficient manner or even synthesized automatically, which can be cost-effectively. In contrast, collecting preference data requires human annotators to evaluate, compare, and rank which can be time-consuming and costly due to the need for careful human judgment.
>
> For the reviewer's question why the computation of AIHF is higher compared to RLHF using preference data onl, it is due to the fact that in our AIHF pipeline, we need to generate data to construct synthetic preference data, then train the reward model and run policy optimization method to finetune the policy. Considering that the algorithm of AIHF has an alternating structure between reward update and policy update, it is reasonable for us to expect that AIHF requires more computation time and achieve better performance.
>
> We also agree with the reviewer's comment that "we need to compare AIHF and RLHF+SFT". Indeed, standard RLHF pipeline has already included SFT step. Actually all our RLHF starts from SFT model, e.g. Fig. 2, 3, and 5 in the revised pdf. Moreover, in Fig. 2 and Fig. 3, we also did ablation study to evaluate AIHF initialized from base model and SFT model. It shows that both can outperform RLHF and a high-quality SFT model can further boost the performance of AIHF.
>
> > 2. In Section 3.4, it is shown that AIHF policy is somehow a weighted average of IRL policy from demonstrations and RLHF policy from preferences, therefore the AIHF policy reduces variance. I agree that this average can reduce variance. Suppose the demonstration needed for SFT is the same demonstration data in AIHF, the standard RLHF has a KL regularization $D_\text{KL}(\pi||\pi_\text{SFT})$ (which relates the learned policy π to the policy $\pi_\text{SFT}$  learned from demonstration data). If we linearize the KL regularization, this may also lead to a (weight) average of  $\pi_\text{SFT}$ (demonstration) and the RLHF policy learned from preference?
>
> **Response**: We thank the reviewer for the question. We updated the paper and added more discussion on the policy induced by SFT+RLHF and by AIHF. In particular, equation (33) and (36) in the updated pdf showed the explicit formula of these two policies, and we showed that RLHF with KL divergence leads to a policy of the form $\pi^*_i(R_D+R_P) = \text{softmax}((R_D+R_P)/\beta)$. If we linearize the KL divergence, the regularization of the KL divergence will not work and then the policy optimization problem turns to be standard reward maximization problem without regularization, which does not have closed form expression. For higher order approximation to the KL divergence, we refer the reviewer to the blog (http://joschu.net/blog/kl-approx.html) which elaborates the details about how to approximate the KL divergence.

---

> > ### Comment · Reviewer_1nbf · 2024-12-02
> >
> > Sorry for the late reply. I appreciate the authors' response. The revision in Section 3.4 is helpful and the authors have done a great job discussing the policy induced by SFT+RLHF. I now have no questions and increase the rating accordingly. I believe that this paper opens a door for future research about IRL for alignment and can contribute to the community.

---

> ### Author Response · Authors · 2024-12-03
>
> Thank you for recognizing the contributions of this work! We truly appreciate your detailed comments and review!

---

### Official Review · Reviewer_Ewdq · 2024-10-28

**Soundness:** 3
**Presentation:** 3
**Contribution:** 3
**Rating:** 6
**Confidence:** 5

**Summary:**

This paper addresses the limitations of current alignment methods, particularly highlighting that the reward model may not be sufficiently well-trained and that demonstrations contain additional information valuable for reward models. The authors then propose a joint learning framework for both reward and policy models to mitigate these issues, demonstrating that their approach can outperform existing RLHF solutions.

**Strengths:**

This paper offers comprehensive theoretical proofs and thorough explanations.

**Weaknesses:**

1. Policy used during preference collection: It is unclear which policy is used during this phase. Based on my understanding, the LLM employs the SFT model to gather human feedback. Meanwhile, the proposed approach seems to assume that the preference dataset is pre-existing before training. In other words, please explicitly state which policy is used to generate samples for human feedback and clarify if a pre-existing preference dataset is assumed to be available or it will be generated during the training process.

2. Unbalanced data claim: The claim that the proposed method performs better with unbalanced data is questionable. Typically, data preprocessing can effectively address this issue. Please provide more evidence supporting this claim, such as by comparing the method to baseline approaches that employ standard data preprocessing techniques for handling imbalanced datasets.

3. Reward model improvement assumption: The framework assumes that incorporating demonstrations leads to a better-trained reward model. However, the paper lacks direct evidence showing that the reward model improves as a result.

4. Effect of human feedback vs. demonstrations: In typical LLM training, the number of demonstrations is orders of magnitude larger than the human feedback dataset. The reviewer is concerned that, during joint training of the reward and policy models, the influence of human feedback might be diminished. Please discuss strategies for balancing the influence of demonstrations and human feedback during joint training, or provide experiments that illustrate the relative impact of each data source on the final model performance.

There are also a few minor issues with the paper that need attention:

(1). The notation for human feedback data in line 099 is identical to the notation used for the trajectory in line 088, which may cause confusion.

(2). Line 232 defines V_theta, but it does not seem to be utilized anywhere in the rest of the paper (except appendix).

**Questions:**

1. The joint training is implemented using a shared parameter θ for both the policy and reward models, which is somewhat unclear. Would it be possible to decouple this into two independent parameters for each model, or are the two models intended to share parameters entirely? This distinction needs further clarification.

2. Figure 2 presents a performance comparison of Pythia models with varying parameter sizes. As the number of parameters increases, the performance gap appears to narrow. Is there any further analysis or explanation provided for this observation?

---

> ### Author Response · Authors · 2024-11-23
>
> We thank the reviewer for the detailed review and we address the reviewer's comments in a point-by-point manner.
>
> **Weaknesses**
> > 1. Policy used during preference collection: It is unclear which policy is used during this phase. Based on my understanding, the LLM employs the SFT model to gather human feedback. Meanwhile, the proposed approach seems to assume that the preference dataset is pre-existing before training. In other words, please explicitly state which policy is used to generate samples for human feedback and clarify if a pre-existing preference dataset is assumed to be available or it will be generated during the training process.
>
> **Response**: In standard RLHF pipeline used in industry, in additional to standard pre-collected preference data, preference data  generated from SFT model and labeled by human labelers is also used. This way, the preference data is made 'on policy', so the trained reward model and the PPO algorithm can better improve upon the SFT.  However in research community, people usually use pre-collected and public preference datasets, due to the expensive human labelling process (it is quite hard to collect a set of 'model-specific' preference data for each popular base SFT model out there).  We have already explicitly provided all the preference data sources with citations, and these are all standard datasets widely used in research cummnity. For example, the Anthropic-HH dataset is a widely used dataset collected from a 52b model, see [1] Section 2.2. We hope this address the reviewer's comment.
>
> > 2. Unbalanced data claim: The claim that the proposed method performs better with unbalanced data is questionable. Typically, data preprocessing can effectively address this issue. Please provide more evidence supporting this claim, such as by comparing the method to baseline approaches that employ standard data preprocessing techniques for handling imbalanced datasets.
>
> **Response**: We thank the reviewer for the comment. We believe that there are some misunderstanding of terminologies here. "unbalanced data" could be referring to unbalanced labels (e.g., the size of the data with certain class are larger than the rest), and we believe the reviewer is referring to this situation. However here we are not referring to unbalanced labels; instead we are referring to the situation where data in different categories are unbalanced (i.e., the size of the demonstration data is larger than that of the preference data). Notice that demonstration data are usually used for SFT and preference data are usually used for reward learning in RLHF pipeline. These two data categories are usually not used together, thus there is no standard preprocessing technique available to make them 'balanced'. This comment is also related to the 4th comment below, and we provide more details of how we balance the influence of the two datasets below.
>
> Finally, we do want to point out that indeed many public models are trained in the 'unbalanced data' regime; see the table we listed in the overall response.
>
> > 3. Reward model improvement assumption: The framework assumes that incorporating demonstrations leads to a better-trained reward model. However, the paper lacks direct evidence showing that the reward model improves as a result.
>
> **Response**: We appreciate this insightfull comment. We conduct extra experiment to show the improvement on the reward model. In particular, we train a reward model using the proposed AIHF altorighm, initiated from mistral-7b-sft-full, and evaluate it using the widely used Reward-Bench [2] to evaluate the reward quality. The result is shown in the followign table (also Tab. 4 in the updated pdf)
> | Reward Model | Chat | Chat Hard | Safety | Reasoning | Average |
> | -------- | ------- | ------- | ------- | ------- | ------- |
> | DPO RM | 37.43% | 55.92% |64.14% | 47.33% | 51.21% |
> | BTL RM | 95.11% | 56.58% | 63.69% | 69.22% | 71.15% |
> | AIHF RM | 94.41% | 55.37% | 63.98% | 76.75% | 72.63% |
>
> In the above table, the DPO reward model is obtained by training a DPO loss on preference data (UltraFeedback) and construct the reward by the log likelihood $r=\log\frac{\pi}{\pi_0}$ where $\pi_0$ is the base model. Note that the reward induced by DPO is known to be generally inferior to explicit reward model due to the limited generalization capability according to [3]. To show the effectiveness of the reward trained by AIHF, which uses both demonstration and preference data, we use the demosntration data UltraChat and generate the non-preferred sample using the base model (mistral-7b-sft-full), then train a reward with UltraFeedback and UltraChat data combined. Note that this way of reward training corresponding to the "Estimating Gradient" in the AIHF update in Algorithm 1.
>
> It can be seen that the reward model estimated by AIHF which can incorporate demonstrations in reward training achieves better performance, especially on the reasoning tasks.

---

> ### Author Response · Authors · 2024-11-23
>
> > 4. Effect of human feedback vs. demonstrations: In typical LLM training, the number of demonstrations is orders of magnitude larger than the human feedback dataset. The reviewer is concerned that, during joint training of the reward and policy models, the influence of human feedback might be diminished. Please discuss strategies for balancing the influence of demonstrations and human feedback during joint training, or provide experiments that illustrate the relative impact of each data source on the final model performance.
>
> **Response**: We again appreciate this insightfull comment. We have taken the strategy for data balancing into consideration, see the weight w_1 in formulation (5). We also include experiment to demonstrate the effect of the data balancing weight in 7B LLM Fine-tuning experiment (AIHF-DPO, different w), as shown in Tab. 6 of the revised pdf. Note that for the experiment in Tab. 6 we use 61.1k preference data and 208k demonstration data, which is exactly an "data unbalance" situation. We hope that these experiments address the reviewer's question about how well the proposed algorithm is able to address the data unbalance situation.
>
>
> Finally, we would like to point that the statement that  "demonstration is order of magnitude large" is not always true. From the table in the general rebuttal, we see that for zephyr-7b-beta model, demonstration:preference  $\approx$ 3.3, for SmolLM2-1.7B-Instruct model, demonstration:preference  $\approx$ 20, and for starchat2-15b-v0.1 model, demonstration:preference  $\approx$ 4.7.
>
>
>
>
> **Minor issues**
>
> > (1). The notation for human feedback data in line 099 is identical to the notation used for the trajectory in line 088, which may cause confusion. (2). Line 232 defines V_theta, but it does not seem to be utilized anywhere in the rest of the paper (except appendix).
>
> **Response**: For (1), actually on line 92 of the original paper, we explained that we actually model human demosntration and preference data as trajectories, so these notations are meant to be identical, that is, $\tau=(s,a)$ where $s$ and $a$ are state and action, is identical to $\tau=(x,y)$ where $x$ is the input prompt and $y$ is the output continuations; For (2), thanks for the reminder, we move it to the appendix; see Appendix A.4.2.
>
> **Questions**
>
> > 1. The joint training is implemented using a shared parameter θ for both the policy and reward models, which is somewhat unclear. Would it be possible to decouple this into two independent parameters for each model, or are the two models intended to share parameters entirely? This distinction needs further clarification.
>
> **Response**: Thanks for the question. We are **not** sharing the parameter for policy and reward. For policy, we have another set of parameter. Note that in the lower level policy optimization, the optimal policy corresponding to the the reward model $R(\cdot;\theta)$ is actually determined by the reward parameter $\theta$, where we can denote the optimal policy under the the reward model $R(\cdot;\theta)$ as $\pi^*_{R_{\theta}}:=\pi_{\theta}$. Therefore we simply put $\pi_{\theta}$ since the optimal policy under a certain reward model $R(\cdot;\theta)$ is actually determined by the reward parameters $\theta$. This is also updated in Sec. 3.1 of the revised pdf.
>
> > 2. Figure 2 presents a performance comparison of Pythia models with varying parameter sizes. As the number of parameters increases, the performance gap appears to narrow. Is there any further analysis or explanation provided for this observation?
>
> **Response**: Thanks for the question. We resize all the plots in Fig. 2 (see the updated pdf) to the same y-scale, and actually for all three model sizes, the gap betwee RLHF and AIHF are consistent. With higher parameters, the average rewards by RLHF and AIHF both increase, but the gaps remain similar (about 0.3-0.5).
>
> References:
>
> [1] Yuntao Bai et al. Training a helpful and harmless assistant with reinforcement learning from human feedback. arXiv preprint arXiv:2204.05862, 2022
>
> [2] Nathan Lambert, Valentina Pyatkin, Jacob Morrison, LJ Miranda, Bill Yuchen Lin, Khyathi Chandu, Nouha Dziri, Sachin Kumar, Tom Zick, Yejin Choi, et al. Rewardbench: Evaluating reward models for language modeling. arXiv preprint arXiv:2403.13787, 2024.
>
> [3] Lin, Yong, et al. "On the limited generalization capability of the implicit reward model induced by direct preference optimization." arXiv preprint arXiv:2409.03650 (2024).

---

> > ### Comment · Reviewer_Ewdq · 2024-11-24
> >
> > Thank you for the authors' detailed explanation and revisions. I have updated my score accordingly.

---

> > > ### Author Response · Authors · 2024-11-24
> > >
> > > We sincerely appreciate your time and effort in reviewing our paper and thank you for recognizing the contributions of this work.

---

### Official Review · Reviewer_mtpH · 2024-11-03

**Soundness:** 3
**Presentation:** 3
**Contribution:** 4
**Rating:** 8
**Confidence:** 3

**Summary:**

The paper proposes a novel learning framework that provides a unified view for reinforcement learning, learning from demonstration, and preference learning. The key idea is to solve all three components simultaneously instead of following the staged approach of typical reinforcement learning with human feedback. The paper discusses many interesting insights, such as framing other algorithms as a special version of the proposed framework, insights into why the proposed method works better, and so on. The proposed work is validated mainly on LLM training problems, but also Mujoco simulated environments are also discussed in the appendix.

**Strengths:**

* The paper presented a novel perspective for preference learning + reinforcement learning problems to approach them simultaneously rather than solving them as separate stages.
* The paper proposed a practical learning algorithm and evaluated it on large-scale LLM data.
* The paper provides numerous interesting insights.

**Weaknesses:**

* In my humble opinion, Section 3.4. WHY AIHF CAN OUTPERFORM TWO-STAGE ALIGNMENT APPROACHES can be improved. Overall, it discusses some mathematical reasons why AIHF works better than RLHF. However, I feel like it depends on several assumptions, such as |D| >> |P|. But is it always true? I always thought preference data was much cheaper than demonstration because it is only a yes/no binary question.
* Also, eventually, RL will dominate, and it can achieve the desirable performance no matter what kinds of data are provided.
* The section is not written concisely compared to its importance. I think it would be better if there was one paragraph that summarized general insights.

**Questions:**

I would appreciate it if the authors could resolve my questions above.

---

> ### Author Response · Authors · 2024-11-23
>
> We thank the reviewer for your positive comments and recognizing the importance of this work. Below, we address the reviewer's comments in a point-by-point manner.
>
> **Weaknesses**
>
> > 1. In my humble opinion, Section 3.4. WHY AIHF CAN OUTPERFORM TWO-STAGE ALIGNMENT APPROACHES can be improved. Overall, it discusses some mathematical reasons why AIHF works better than RLHF. However, I feel like it depends on several assumptions, such as |D| >> |P|. But is it always true? I always thought preference data was much cheaper than demonstration because it is only a yes/no binary question.
>
> **Response**: We thank the reviewer for the comment. Our derivation and reasoning does not depend on assumption |D| >> |P|. We indeed showed that when |D| >> |P|, AIHF can yield better policy estimation than RLHF. Meanwhile, when |D| << |P|, we have similar argument that AIHF yield better policy estimation and we put it in Appendix A.3.2 due to page limits. Essentially, we would argue that AIHF is more capable of handling data imbalance situation. Moreover, the derivation also provide some hint that the AIHF policy has less variance as compared with the RLHF policy (as the former represents a weighted averaged of two policies); see the discussion Appendix A.3.2 for a concrete example.
>
>
> Additionally, we do want to point out that indeed many public models are trained with $|D|\gg |P|$; see the table we listed in the overall response. One potential reason why many public models are trained with more demonstrations is that demonstrations can sometimes be collected from existing datasets or created by experts in a more efficient manner or even synthesized automatically, which can be cost-effectively. In contrast, collecting preference data requires human annotators to evaluate, compare, and rank which can be time-consuming and costly due to the need for careful human judgment.
>
> > 2. Also, eventually, RL will dominate, and it can achieve the desirable performance no matter what kinds of data are provided.
>
> **Response**: We appreciate this insightful comment and we fully agree with the reviewer that RL is a powerful tool. However, currently one bottleneck for RLHF and other LLM alignment pipelines is the data collection for both high quality demonstration and preference data. Therefore it is meaningful to discuss and address the situations when both demonstration and preference data are limited and how to effectively leverage the demonstration and the preference data to further unlock the potential of RL in LLM alignment, and this is exactly the purpose of this paper.
>
> > 3. The section is not written concisely compared to its importance. I think it would be better if there was one paragraph that summarized general insights.
>
> **Response**: Thanks for the suggestion. We have concluded Section 3.4 with a 'discussion' section, which summarizes a few of our observations. We hope that this helps distill our insights.

---

### Official Review · Reviewer_7hXs · 2024-11-04

**Soundness:** 3
**Presentation:** 4
**Contribution:** 3
**Rating:** 5
**Confidence:** 3

**Summary:**

This work introduces the Alignment with Integrated Human Feedback (AIHF) framework for learning a policy from both demonstrations and preferences. AIHF poses reward and policy learning as a single bi-level optimization problem where the outer loop maximizes optimizes the policy to fit the demonstrations and the reward to fit the preferences and the inner optimizes the policy with respect to the learned reward. The paper illustrates how AIHF connects to prior alignment algorithms and proposes a concrete instantiation of AIHF. Empirically, AIHF improves the alignment over standard RLHF.

**Strengths:**

1. The single stage-learning of reward and policy from AIHF learns a more robust reward model that leverages both demonstrations and preferences compared to two stage approaches that first learn a reward function from only preference data.

1. The paper demonstrates how the AIHF framework can be specialized to an RLHF, DPO, or self-play like approach. This shows that AIHF offers a more general alignment formulation.

1. Section 3.4 theoretical and numerical evidence for why AIHF is superior to a two-stage alignment process like in standard RLHF.

1. The paper provides performance guarantees for the proposed AIHF algorithm.

1. AIHF outperforms RLHF and regular SFT on the Anthropic-HH dataset across several Pythia model sizes.

1. AIHF with the DPO and Self-Play instantiation improves the performance of an RLHF model when trained with the Ultrafeedback-binary preference dataset and Ultrachat200k demonstration dataset.

1. Results in the supplementary also show AIHF improves performance relative to RLHF for continuous control tasks.

**Weaknesses:**

1. The paper claims that AIHF outperforms existing alignment methods when the data is unbalanced (L78, L322), but this claim does not appear supported by the results in Section 5. The results in Figure 4 right show AIHF suffering as the preference and demonstrations become unbalanced. Contrary to the caption in Figure 4, these results also do not test if AIHF outperforms RLHF with different demonstration ratios since no RLHF result is displayed in Figure 4 right. This leaves it unclear if AIHF does have any benefit over existing alignment algorithms in unbalanced datasets.

1. The evaluation of AIHF in Section 5 is hard to follow. Figure 2 evaluates the performance of the proposed AIHF algorithm from Section 4, but Figure 3 evaluates the DPO and Self-Play versions. This makes it difficult to evaluate the empirical significance of the AIHF algorithm proposed in Section 4.


1. Insufficient empirical comparison to prior work. The paper does not evaluate the performance of the specialized variants of AIHF against the existing versions of the algorithms such as DPO and SPIN. These comparisons are crucial for evaluating the benefits of the AIHF framework. Additional comparisons to other existing such as IPO [1] would also strengthen the results.

1. The paper does not clearly discuss the limitations of AIHF.

1. The MuJoCo results in Appendix A.2.1 are missing important comparisons. Again, Figure 5 does not compare against existing alignment algorithms like DPO. Additional results comparing the quality and balance of this preference and demonstration data would also strengthen these results. This would confirm if RLHF is indeed suffering due to low-quality preference data as claimed.

Minor:
1. L476 should explicitly reference Figure 4. When first reading, it was unclear where this study was located.

1. Figure 3 should provide exact numbers of the bars in the chart for more detailed empirical comparisons since many of the results are very close.

[1] Calandriello, Daniele, et al. "Human alignment of large language models through online preference optimisation." arXiv preprint arXiv:2403.08635 (2024).

**Questions:**

1. In Section 6, Why not compare all of AIHF, Self-Play-AIHF, and AIHF-DPO in Figures 2 and 3? Does the algorithm proposed in Section 4 empirically outperform AIHF-DPO and Self-Play-AIHF?

1. Are the improvements of AIHF over the bsae Zephyr-Beta model significant? It appears the average improvement is only a couple of percent?

---

> ### Author Response · Authors · 2024-11-23
>
> We thank the reviewer for the detailed review of the paper and the valuable feedback. Below, we address the reviewer's questions in a point-by-point manner.
>
> **Weaknesses**
> > 1. The paper claims that AIHF outperforms existing alignment methods when the data is unbalanced (L78, L322), but this claim does not appear supported by the results in Section 5. The results in Figure 4 right show AIHF suffering as the preference and demonstrations become unbalanced. Contrary to the caption in Figure 4, these results also do not test if AIHF outperforms RLHF with different demonstration ratios since no RLHF result is displayed in Figure 4 right. This leaves it unclear if AIHF does have any benefit over existing alignment algorithms in unbalanced datasets.
>
> **Response**: We thank the reviewer for the question. We partially agree with your statement that the right side of the original Fig. 4 is not pertinent to show that AIHF is good for unbalanced data. Ideally, we need to show that, under different demonstration/preference ratio, AIHF can consistently outperform RLHF. So we updated Fig. 4 accordingly in the revised draft. Note that in the new pdf Fig. 4 is relabelled as Fig. 3. Specifically, the new Fig. 3(b) shows the case when dem:pref = 10,000:5,000 and the new Fig. 3($\text{c}$) is dem:pref = 10,000:10,000. We can see that AIHF consistently outperforms RLHF. We hope this addresses the reviewer's comment.
>
> > 2. The evaluation of AIHF in Section 5 is hard to follow. Figure 2 evaluates the performance of the proposed AIHF algorithm from Section 4, but Figure 3 evaluates the DPO and Self-Play versions. This makes it difficult to evaluate the empirical significance of the AIHF algorithm proposed in Section 4.
>
> **Response**: We thank the reviewer for pointing this out. The reason we didn't exactly do AIHF in Fig. 3 (of the original submission) was that, it was hard to conduct policy optimization methods such as PPO for large 7b models with the computation resrouces that are available to us (i.e., we did not have access to A100 GPUs with 80G memory at the time).
>
> In the revised manuscript, we have implemented AIHF as it was originally described with explicit policy optimization (see Appendix A.2.4 of the revised pdf for the details of the implementation). We updated Fig. 3 by including the  AIHF result (it is now the Fig. 4 in the revised pdf). Further, we have included (in the revised pdf) more results in Fig. 2, Fig. 3 and Tab. 6 under the same experiment setting of Fig. 2. From these new results we can see that policy-optimization based AIHF indeed has an outstanding performance comparing to its variants, namely AIHF-DPO and self-play AIHF for all our experiment settings. We hope that these new experiments sufficiently address reviewer's concern about the advantage of AIHF proposed in Section 4.
>
> > 3. Insufficient empirical comparison to prior work. The paper does not evaluate the performance of the specialized variants of AIHF against the existing versions of the algorithms such as DPO and SPIN. These comparisons are crucial for evaluating the benefits of the AIHF framework. Additional comparisons to other existing such as IPO [1] would also strengthen the results.
>
> **Response**: Thank you for your comment. In the revised manuscript, we have conducted some comprehensive comparison to prior works, such as DPO, SPIN and IPO; see Fig. 4 in the main paper and Tab. 3 and Tab. 5 in the appendix of the updated pdf. All the results suggest that the proposed AIHF still outperforms these SOTA methods.
>
> > 4. The paper does not clearly discuss the limitations of AIHF.
>
> **Response**: We thank the reviewer for pointing this out. We added a paragraph at the end of the revised paper (in conclusion section) to discuss the limitation of AIHF.
>
> > 5. The MuJoCo results in Appendix A.2.1 are missing important comparisons. Again, Figure 5 does not compare against existing alignment algorithms like DPO. Additional results comparing the quality and balance of this preference and demonstration data would also strengthen these results. This would confirm if RLHF is indeed suffering due to low-quality preference data as claimed.
>
> **Response**: Thank you for your suggestion. We have added the comparison between AIHF and a DPO-type algorithm in Fig. 5. In particular, as the authors claimed in [1], the Inverse Preference Learning (IPL) algorithm is actually the DPO-type algorithm applied to multi-horizon MDP, therefore it is an ideal baseline for the MuJoCo setting (since it the underlying problem is a multi-horizon MDP). We can see that AIHF outperforms this algorithm. We also conducted additional experiments that compare different settings where we vary the quality and balance of preference and demonstration data; see Fig. 3 and Tab. 5 in the revised pdf. Once again, the proposed AIHF still outperforms this new baseline.

---

> ### Author Response · Authors · 2024-11-23
>
> **Minor**
> > 1. L476 should explicitly reference Figure 4. When first reading, it was unclear where this study was located.
>
> **Response**: Thanks, we updated it in the revised pdf.
>
> > 2. Figure 3 should provide exact numbers of the bars in the chart for more detailed empirical comparisons since many of the results are very close.
>
> **Response**: We thank the reviewer for the suggestion. We include these numbers in Table 3 in the Appendix (in both original and the updated pdf).
>
> **Questions**
>
> > 1. In Section 6, Why not compare all of AIHF, Self-Play-AIHF, and AIHF-DPO in Figures 2 and 3? Does the algorithm proposed in Section 4 empirically outperform AIHF-DPO and Self-Play-AIHF?
>
> **Response**: This is related to the comment in the weaknesses. In Figure 3 of the revised pdf, we add extra comparison with AIHF-DPO and Self-Play-AIHF. We also updated Fig. 4 in the revised pdf to show a comprehensive comparison between AIHF, Self-Play-AIHF, and AIHF-DPO. Our implementation of the Algorithm 1 indeed outperforms AIHF-DPO and Self-Play-AIHF.
>
> > 2. Are the improvements of AIHF over the base Zephyr-Beta model significant? It appears the average improvement is only a couple of percent?
>
> **Response**: Yes, we do believe that making improvement for about 3% from zephyr-7b-beta (see exact numbers in Tab. 3) is significant. The Open LLM leaderboard benchmark is one of the most heavily-used (SPIN [2] also primarily used this) influential benchmark in the community where it does not rely on large models such as GPT4 but only on multiple downstream tasks to test the performance of different LLMs. For models like Llama3-70B, they in general also only achieve 1% to 2% increase over the previous state-of-the-art (if you check their [hugginface webpage](https://huggingface.co/open-llm-leaderboard), they also tested on tasks such as MMLU, GSM8K and Winogrande). Therefore we believe we are making meaningful progress using the same tasks as SPIN [2] on 7b models.
>
> References:
>
> [1] Joey Hejna and Dorsa Sadigh. Inverse preference learning: Preference-based rl without a reward function. arXiv preprint arXiv:2305.15363, 2023
>
> [2] Chen, Zixiang, et al. "Self-play fine-tuning converts weak language models to strong language models." International Conference on Machine Learning (2024).

---

> ### Author Response · Authors · 2024-11-25
> **Looking Forward to Post-Rebuttal Feedback**
>
> Dear reviewer 7hXs,
>
> Thank you very much for taking the time to review our paper! We cherish your comments and evaluations very much! In our posted responses, we have made a point-to-point response to alleviate your concerns. If you have any further concerns on our response, we are more than happy to address them.
>
> Best,
> Authors

---

### Author Response · Authors · 2024-11-23
**Official comment for all reviewers**

We thank all the reviewers for constructive comments and suggestions. We upload a revised pdf draft, where we have done major revision over the experiment section (Section 5) and add extensive experiments as well as ablation study to address the concerns from reviewers. Major revised parts are highlighted in blue.

Since our paper is focusing on the situation where the demonstration and preference data are unbalanced, we include several public models and amount of demonstration and preference data they use for alignment:
| Model Name | Demonstration Data | Preference Data |
| -------- | ------- | ------- |
| HuggingFaceH4/zephyr-7b-beta | HuggingFaceH4/ultrachat_200k (200k) | HuggingFaceH4/ultrafeedback (60k) |
| HuggingFaceTB/SmolLM2-1.7B-Instruct | HuggingFaceTB/smoltalk (1.1M) | HuggingFaceH4/ultrafeedback (60k) |
|HuggingFaceH4/starchat2-15b-v0.1| HuggingFaceH4/airoboros-3.2 (58k) + HuggingFaceH4/Code-Feedback (65k) + HuggingFaceH4/orca-math-word-problems-200k (200k) + HuggingFaceH4/SystemChat (6.5k) + HuggingFaceH4/capybara (16k)  | HuggingFaceH4/ultrafeedback_binarized (60k) + HuggingFaceH4/orca_dpo_pairs (12.4k) |

From the above table, we indeed observe that typically the size of the demonstration data is larger than the preference data, therefore the  data 'unbalanced' situation often arises.

For our two experiment setting in Sec. 5, we include the numbers of demonstration and preference data in the following table.
| Experiment setting | Demonstration Data | Preference Data |
| -------- | ------- | ------- |
| 1b/2.8b | 10k | 5k/10k |
| 7b | 208k | 61.1k |

Note that for our 7b experiment setting, the data is always unbalanced and the demonstration dataset is significantly larger than the preferece.

We include the point-to-point response to all reviewers below.

---

### Meta-Review · Area_Chair_69id · 2024-12-23

**Metareview:**

This paper introduces a novel algorithmic framework that effectively integrates expert demonstrations and pairwise comparisons from human feedback to learn reward functions. It tackles the critical issue of aligning AI models with human preferences, which is essential for the safe deployment of AI systems in real-world applications.

While the paper presents results from controlled experiments, there may be a lack of real-world application scenarios that could validate the effectiveness of the proposed approach in practical settings

The paper is well written and easy to follow. The AC agrees with the majority of the reviewers that it should be accepted by ICLR 2025.

**Additional Comments On Reviewer Discussion:**

After discussion, one reviewer raised the score from 8 to 10, one reviewer raised the score from 5 to 6. The additional info provided by the authors confirmed the majority of the authors.

---

### Decision · Program_Chairs · 2025-01-22

Accept (Spotlight)